# Laser scanning reflection-matrix microscopy for aberration-free imaging through intact mouse skull

Seokchan Yoon [1,2,4], Hojun Lee[1,2,4], Jin Hee Hong[1,2], Yong-Sik Lim[3] & Wonshik Choi [1,2✉]

A mouse skull is a barrier for high-resolution optical imaging because its thick and inhomogeneous internal structures induce complex aberrations varying drastically from position to position. Invasive procedures creating either thinned-skull or open-skull windows are often required for the microscopic imaging of brain tissues underneath. Here, we propose a label-free imaging modality termed laser scanning reflection-matrix microscopy for recording the amplitude and phase maps of reflected waves at non-confocal points as well as confocal points. The proposed method enables us to find and computationally correct up to 10,000 angular modes of aberrations varying at every $10 \times 10\ \mu m^2$ patch in the sample plane. We realized reflectance imaging of myelinated axons in vivo underneath an intact mouse skull, with an ideal diffraction-limited spatial resolution of 450 nm. Furthermore, we demonstrated through-skull two-photon fluorescence imaging of neuronal dendrites and their spines by physically correcting the aberrations identified from the reflection matrix.

[1] Center for Molecular Spectroscopy and Dynamics, Institute for Basic Science, Seoul 02841, Korea. [2] Department of Physics, Korea University, Seoul 02855, Korea. [3] Department of Nano Science and Mechanical Engineering and Nanotechnology Research Center, Konkuk University, Chungbuk, Korea. [4]These authors contributed equally: Seokchan Yoon, Hojun Lee. ✉email: wonshik@korea.ac.kr

Confocal imaging is one of the most widely used configurations in deep-tissue optical microscopy due to its ability to reject signals from out-of-focus planes and other unwanted noise by multiple light scattering[1,2]. The technique has been employed either by itself in confocal reflectance/fluorescence imaging[3] or combined with temporal gating to further attenuate multiple-scattering noise[4]. In the presence of sample-induced aberrations, however, signals containing object information are spread away from the confocal pinhole by the blur of the point-spread function (PSF)[5–8]. Image contrast is thereby reduced, and resolving power is gradually lost with increasing imaging depth. The key to resolving sample-induced aberrations and achieving ideal diffraction-limited imaging deep within a scattering medium is to coherently refocus the non-confocal signals, or signals arriving at positions other than the confocal pinhole, back to the confocal detection position. Since multiple-scattering noise arrives at non-confocal positions as well as spatially aberrated signals, it is critical to selectively refocus the aberrated signals, but not the multiple-scattering noise. Up to this point, selective refocus has been extremely difficult in label-free reflectance imaging compared with fluorescence imaging, mainly because the signal and multiple-scattering noise have identical optical frequencies.

The most straightforward approach for focusing non-confocal signals back to the confocal position is to place a wavefront shaping device such as a deformable mirror or liquid-crystal spatial light modulator (SLM) in the illumination and/or detection beam paths. Selective focusing of the aberrated signal is achieved by adjusting the SLM iteratively to maximize the intensity or sharpness of the resulting confocal image[9–13]. These so-called sensorless adaptive optics (AO) methods can identify high-order aberrations, mainly in fluorescence imaging. However, the feedback process is time-intensive because an image acquisition is required for each iteration step. A model-based, modal aberration correction approach might reduce optimization time, but only works on the lowest modes of Zernike polynomials[14]. Furthermore, the iterative optimization can only be initiated when the objects are visible in the first place.

Various approaches have been proposed based on direct wavefront measurements to correct aberrations with the minimal number of image acquisitions; these are referred to as wavefront-sensing AO. The Shark–Hartmann wavefront sensor[15–19] and interference microscopy[20–23] have been widely used for wavefront recording. In general, sample-induced aberrations in both the illumination and detection beam paths contribute to the measured wavefront. Mathematically, the detection PSF is convolved with the multiplication of the object function and illumination PSF, and this makes it difficult to extract individual PSFs. Previous studies have resolved this double-pass problem in limiting cases when there exist bright, point-like scatterers within the sample, similar to guide stars in astronomy[6]. This changes the multiplication of the object function and illumination PSF to a point source regardless of the distortion of the illumination PSF. However, guide stars are available only in special cases. For example, photoreceptor cells in retina imaging[18] and nonlinear excitation can serve as intrinsic and approximate guide stars[15,16,19,24,25]. Another strategy is to illuminate the planar wave, but not the focused wave, to minimize aberrations in the illumination beam path. Aberrations are then obtained from a reflected wavefront by computationally optimizing image metrics[22,23,26]. The drawback of this approach is its susceptibility to multiple-scattering noise due to the loss of confocal gating. These limitations of existing wavefront-sensing AO approaches are mainly due to incomplete recording of the input-output response of the specimens, which makes the double-pass problem underdetermined.

We hereby propose a laser-scanning reflection-matrix microscopy (LS-RMM) method that records both non-confocal and confocal signals of elastic backscattering from the sample in their phases and amplitudes. Low-coherence interferometry is employed for time-gated detection of the backscattered waves; this excludes multiple-scattering noise arriving at different flight times from the signal waves. Together, these measurements constitute a reflection matrix that quantifies the complete input-output response of the light-medium interaction. Conventional optical coherence microscopy (OCM) combining confocal microscopy and optical coherence tomography measures a subset of the reflection matrix. We take advantage of non-confocal signals to unambiguously identify one-way aberrations from highly complex round-trip distortions. The processing step is the transformation of the measured reflection matrix taken for focused illuminations to that for planar illuminations. We then administered a unique algorithm referred to as the closed-loop accumulation of single scattering (CLASS), developed previously to selectively identify and computationally correct sample-induced aberrations from background multiple-scattering noise[27–29]. In particular, we improved the algorithm to correct a large number of correction modes (~10,000 angular modes) at each local area down to $10 \times 10 \ \mu m^2$. By using this capacity for correcting local high-order aberrations, we demonstrated label-free, in vivo imaging of myelinated axons underneath an intact mouse skull, an extreme form of aberration medium[30]. Furthermore, by displaying the conjugation of the identified aberration maps on a spatial light modulator in the excitation beam path, we performed two-photon fluorescence (TPF) imaging of the neuronal dendrites and visualized their spines with the spatial resolution of 500 nm, close to the diffraction-limited TPF imaging resolution of 380 nm, over the field of view much larger than the isoplanatic patch.

## Results

**Experimental schematic of laser-scanning reflection-matrix microscopy.** The LS-RMM is built on the same backbone as the OCM system, but with a difference in the detection scheme. A camera was placed on a plane conjugate to the image plane (instead of the confocal pinhole and a photodetector), and a reference wave was introduced to the camera to form off-axis low-coherence interferometry (Fig. 1a, "Methods", and Supplementary Note 1). The time-gated electric-field (E-field) images for waves backscattered from the sample were then retrieved from the interferogram (Fig. 1b–e and Supplementary Note 2). In the absence of aberrations, the obtained E-field image was sharply focused with a diffraction-limited spot size of 450 nm (intensity map, Fig. 1b; phase map, Fig. 1c). On the contrary, the intensity map taken in the presence of an aberrating medium showed a significantly broadened distribution with the peak intensity attenuated by almost two orders of magnitude (Fig. 1d). The corresponding phase map also showed a broadened distribution of meaningful phase values (Fig. 1e).

Image acquisition was conducted using the focused illumination beam scanned in raster mode by the two galvanometer mirrors (GMs) over a given field of illumination (FOI) area on the sample plane with a diffraction-limited scanning interval of $\lambda/2\alpha$, where $\lambda$ is the illumination wavelength, and $\alpha$ is the numerical aperture (NA) of the objective lens. The FOI is equivalent to the field of view of the conventional OCM. For each illumination point $\mathbf{r}_i$ at the sample plane, the E-field image $E_{cam}(\mathbf{r}_{cam}; \mathbf{r}_i)$ was recorded in the de-scanned camera coordinate frame with the position vector $\mathbf{r}_{cam}$. In contrast to previous wavefront-sensing AO, where a wavefront detector was located at a conjugate pupil plane[20,31], the present scheme offers the best spatial distinction

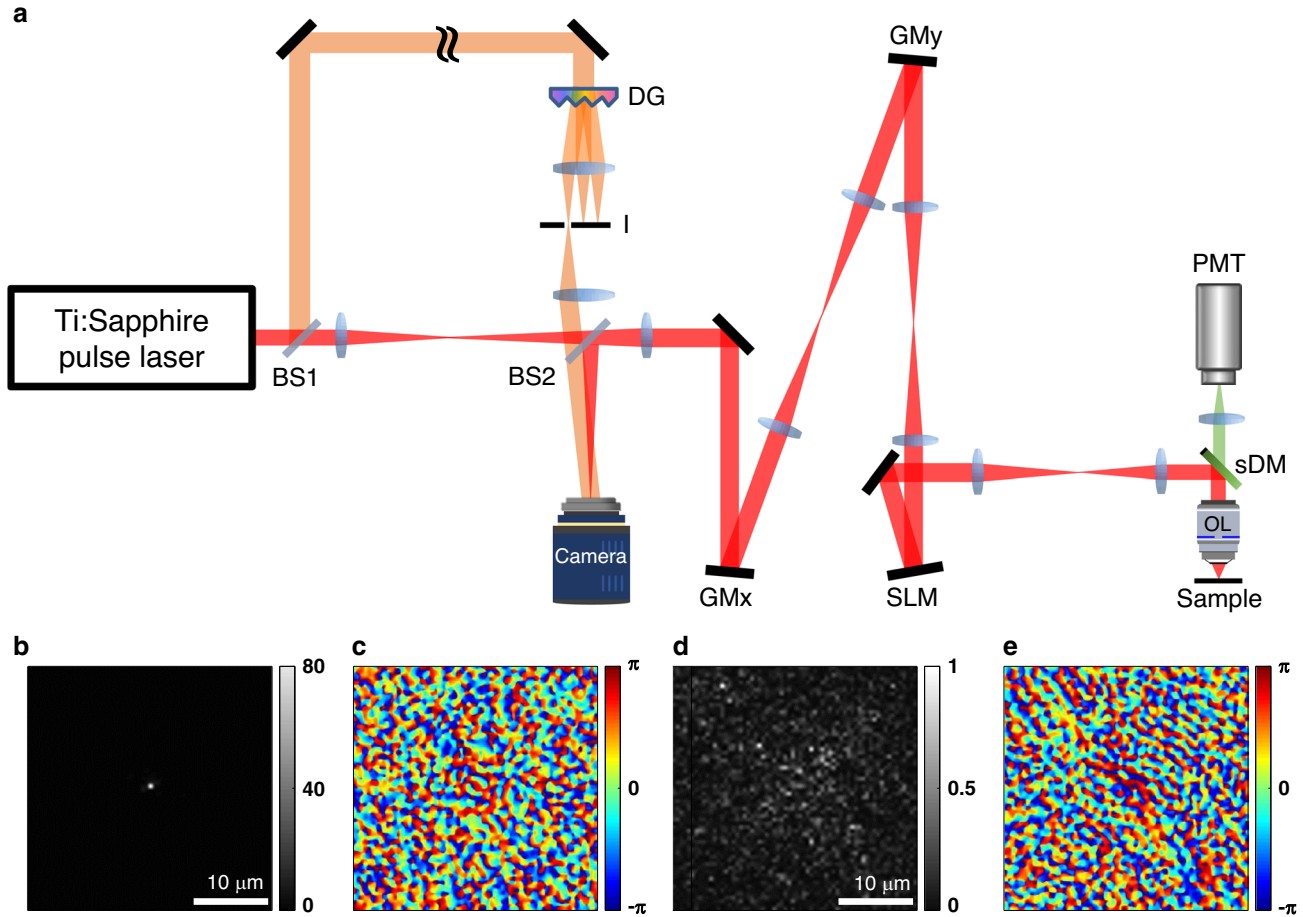

**Fig. 1 Experimental schematic of laser-scanning reflection-matrix microscopy. a** Experimental schematic; BS1 and BS2 beam splitters. The sample and reference beams are colored in red and orange, respectively, for visibility. GMx and GMy, galvanometer mirrors for scanning the sample beam along $x$ and $y$ axes, respectively. SLM spatial light modulator, sDM short-pass dichroic mirror, OL objective lens, PMT photomultiplier tube, DG diffraction grating, P pinhole. **b**, **c** Intensity and phase images of the reflected wave $E_{cam}(\mathbf{r}_{cam}; \mathbf{r}_i)$, from the sample, respectively, for samples without aberrations. **d**, **e** Same as **b** and **c**, but for an aberrating sample. Color bars in **b** and **d** indicate intensity normalized by the maximum intensity in **d**. Color bars in **c** and **e** indicate phase in radians. Scale bars, 10 μm.

between the signal and multiple-scattering noise[32]. Here, the detection area at the camera, called the field of detection (FOD), was adjusted to be wide enough to capture the entire profile of the reflected wave broadened by the sample-induced aberrations. The number of correction modes $N_c$ is determined by the number of free modes within the FOD. As discussed in detail in "Methods", in a typical FOD of $50 \times 50$ μm², we can correct up to 10,000 angular modes of aberrations in the pupil. The choice of optimal sizes for the FOD and FOI is made in consideration of PSF broadening, the intensity ratio between single- and multiple-scattered waves, and image acquisition time. The wider the FOD used, the higher order of aberrations can be measured and corrected, with a concomitant increase in data acquisition time (Supplementary Note 5), and the more multiple-scattering noise is captured. We demonstrated the hardware correction of the identified aberration by inserting an SLM (X10568-02, Hamamatsu) at the plane conjugate to the objective lens in the illumination beam path. The SLM serves as a flat mirror for reflection matrix recording and image acquisition of label-free reflectance imaging. The device was used for multi-photon imaging to physically compensate for the sample-induced aberrations identified by the CLASS algorithm.

The LS-RMM proposed here is different from previous reflection-matrix-based approaches in its illumination and detection configurations[27,28,33]. Our earlier studies used planar wave illuminations, while the present LS-RMM uses a focused illumination. The main benefit therein is the enhanced signal to multiple-scattering noise ratio. In planar wave illumination, multiple-scattering noise generated by illumination over a wide area is summed at each detection pixel. On the contrary, signals generated by a focused illumination compete only with the multiple-scattering noise generated by the given focused illumination itself[32]. Other approaches employing focused illumination usually put a detector in the pupil plane where signals are spread out over the detection area[20,31,34]. In contrast, we placed a wavefront detector at the image plane where signals are more likely to be focused relative to multiple-scattering noise, providing an optimal spatial distinction between the signal and multiple-scattering noise. This technique ensures the optimal use of the detector's dynamic range, resulting in enhanced sensitivity to multiple-scattering noise (Supplementary Note 4). Furthermore, the matrix acquisition time can be shortened in the new configuration. The view field for recording non-confocal signals needs to be wide enough to capture the entire PSF broadened by the sample-induced aberrations. We can adaptively adjust this view field at the camera with the increase of imaging depth in such a way to optimize matrix acquisition speed. This is especially critical for in vivo imaging where the motion of the living specimen can undermine the image reconstruction process.

Another important benefit of focused illumination geometry lies in its direct compatibility with existing imaging modalities such as OCM and multi-photon microscopy (MPM). The AO system can be integrated with these conventional imaging modalities by inserting wide-field, low-coherence interferometry in the detection plane, allowing the combination of hardware-adaptive optics (HAO) and computational adaptive optics (CAO)[21] in complement to one another. We realized HAO by displaying the conjugation of the aberration maps identified from the reflection matrix by simply inserting the SLM in the illumination beam path and demonstrated AO multi-photon imaging through an intact mouse skull. We would like to emphasize that the realization of LS-RMM in a focused illumination configuration has been technically challenging due to shot-to-shot phase fluctuations during scanning, especially for in vivo imaging applications. We maintained phase-referencing between different illuminations by improving the mechanical stability of the interferometry and laser-scanning system.

**Image processing and aberration correction**. In the case of optical coherence reflectance imaging, sample-induced aberrations of the incident wave on the way to the sample and those of the reflected wave on its way back to the detector must be separately identified. This necessitates the recording of a reflection matrix and the application of the CLASS algorithm. In this section, we described how the sequence of images taken by LS-RMM was processed to construct a time-gated reflection matrix in the space domain. For a given focused illumination at a position $\mathbf{r}_i$ in the sample plane, the time-gated $E$-field of the reflected wave $E_{cam}(\mathbf{r}_{cam}; \mathbf{r}_i)$ recorded at the position $\mathbf{r}_{cam}$ in the camera plane is as follows (see Supplementary Note 3 for the coordinate system):

$$E_{cam}(\mathbf{r}_{cam}; \mathbf{r}_i) = \int P_o(\mathbf{r}_{cam}; \mathbf{r}) \cdot [O(\mathbf{r}) \cdot P_i(\mathbf{r}; \mathbf{r}_i)] d^2\mathbf{r} + E_M(\mathbf{r}_{cam}; \mathbf{r}_i). \tag{1}$$

Here, $P_i(\mathbf{r}; \mathbf{r}_i)$ is the illumination $E$-field PSF at a position $\mathbf{r}$ in the sample plane, which is broadened with respect to $\mathbf{r}_i$ due to sample-induced aberrations. $O(\mathbf{r})$ is the object function given by the amplitude reflectance of the target object. $P_o(\mathbf{r}_{cam}; \mathbf{r}_i)$ describes the detection $E$-field PSF of the returning waves from the sample plane to the camera plane. $E_M(\mathbf{r}_{cam}; \mathbf{r}_i)$ represents a speckle field from the time-gated multiple-scattered waves by the scattering layer located above the sample plane. Multiple-scattered waves whose flight times are either longer or shorter than that of the signal wave are rejected by the time-gated detection.

We next needed to obtain the $E$-field map of the reflected wave $E_{lab}(\mathbf{r}_o; \mathbf{r}_i)$ in the laboratory frame coordinate $\mathbf{r}_o$ to identify the sample-induced aberrations. Since $E_{cam}(\mathbf{r}_{cam}; \mathbf{r}_i)$ is acquired after the de-scanning action by the GMs, $\mathbf{r}_{cam}$ and $\mathbf{r}_o$ have the relation $\mathbf{r}_{cam} = \mathbf{r}_o - \mathbf{r}_i$. The $E$-field in the laboratory frame can thereby be obtained from the relation $E_{lab}(\mathbf{r}_o; \mathbf{r}_i) = E_{cam}(\mathbf{r}_o - \mathbf{r}_i; \mathbf{r}_i)$. For demonstrating the working principle of the LS-RMM system, we performed reflectance imaging of a custom-made Siemens star target placed under a 600-μm-thick, rough-surfaced plastic layer exhibiting strong aberrations (Fig. 2a). A set of $E$-field images $E_{lab}(\mathbf{r}_o; \mathbf{r}_i)$ for all illumination spots $\mathbf{r}_i$ within an FOI of $40 \times 40$ μm² was recorded (Fig. 2b). A time-gated reflection matrix $\mathbf{R}(\mathbf{r}_o; \mathbf{r}_i)$ in the space domain was constructed by assigning $E_{lab}(\mathbf{r}_o; \mathbf{r}_i)$ to each element of $\mathbf{R}$ in such a way that the $\mathbf{r}_i$ and $\mathbf{r}_o$ are the column and row indices of $\mathbf{R}$, respectively (Fig. 2c). The acquired reflection matrix $\mathbf{R}$ is represented as

$$\mathbf{R} = \mathbf{P}_o \mathbf{O} \mathbf{P}_i + \mathbf{M}. \tag{2}$$

Here, $\mathbf{O}$ is a diagonal matrix containing the target's object function in the diagonal elements; $\mathbf{O}(\mathbf{r}; \mathbf{r}) = O(\mathbf{r})$. $\mathbf{P}_i$ is the illumination PSF matrix whose element is given by $P_i(\mathbf{r}; \mathbf{r}_i)$. The detection PSF matrix $\mathbf{P}_o$ consists of $P_o(\mathbf{r}_o; \mathbf{r})$, which is theoretically identical to the transpose of $\mathbf{P}_i$ in epi-detection geometry but can differ in the presence of a slight mismatch between illumination and detection optics. $\mathbf{M}$ is the multiple-scattering matrix composed of $E_M(\mathbf{r}_o; \mathbf{r}_i)$. The conventional OCM image can be obtained from the main diagonal elements of $\mathbf{R}$ as they correspond to the confocal signals (Fig. 2d). The OCM image was blurred and severely distorted due to pronounced sample-induced aberrations contained in $\mathbf{P}_i$ and $\mathbf{P}_o$. We then identified and removed $\mathbf{P}_i$ and $\mathbf{P}_o$ in $\mathbf{R}$ to reconstruct an aberration-free object image.

To apply the CLASS algorithm, we converted the position basis of the reflection matrix $\mathbf{R}$ into the spatial frequency basis ($\mathbf{k}$ space) by applying a Fourier transform operator $\mathbf{F}$ and the inverse operator $\mathbf{F}^{-1}$ to the output and input sides of $\mathbf{R}$, respectively: $\tilde{\mathbf{R}} = \mathbf{F}\mathbf{R}\mathbf{F}^{-1}$. The reflection matrix $\tilde{\mathbf{R}}$ on the spatial frequency basis (Fig. 2e) can be expressed as

$$\tilde{\mathbf{R}} = \tilde{\mathbf{P}}_o \tilde{\mathbf{O}} \tilde{\mathbf{P}}_i + \tilde{\mathbf{M}}. \tag{3}$$

Here, $\tilde{\mathbf{O}} = \mathbf{F}\mathbf{O}\mathbf{F}^{-1}$ is a circulant matrix whose elements are given by the spatial frequency spectrum of the object function $\tilde{O}(\mathbf{k})$: $\tilde{\mathbf{O}}(\mathbf{k}_o; \mathbf{k}_i) = \tilde{O}(\mathbf{k}_o - \mathbf{k}_i)$ where, $\mathbf{k}_i$ and $\mathbf{k}_o$ are the input and output transverse wavevectors, respectively. $\tilde{\mathbf{P}}_i = \mathbf{F}\mathbf{P}_i\mathbf{F}^{-1}$ and $\tilde{\mathbf{P}}_o = \mathbf{F}\mathbf{P}_o\mathbf{F}^{-1}$ are the transmission matrices for a plane wave propagating through the illumination and detection pathways, respectively. For the area where the PSF is shift-invariant, which is the case within an isoplanatic patch, the relation $P_i(\mathbf{r}; \mathbf{r}_i) = P_i(\mathbf{r} - \mathbf{r}_i)$ is valid. Then, $\mathbf{P}_i$ takes the form of a Toeplitz matrix, which sets $\tilde{\mathbf{P}}_i$ as a diagonal matrix whose diagonal element consists of $\exp[i\phi_i(\mathbf{k}_i)]$. Likewise, $\tilde{\mathbf{P}}_o$ is a diagonal matrix with the diagonal element given by $\exp[i\phi_o(\mathbf{k}_o)]$. Here, $\phi_i(\mathbf{k}_i)$ and $\phi_o(\mathbf{k}_o)$ are angle-dependent phase retardations induced by the scattering layer for the illumination and detection pathways, respectively.

We then applied the CLASS algorithm[27,28] to $\tilde{\mathbf{R}}$, which separately identifies $\phi_i(\mathbf{k}_i)$ and $\phi_o(\mathbf{k}_o)$ such that the total intensity of the confocal image, reconstructed from an aberration-corrected reflection matrix, is maximized (see "Methods"). The conjugations of the pupil functions were applied to the original matrix $\tilde{\mathbf{R}}$ to obtain the aberration-corrected matrix, $\tilde{\mathbf{R}}_c = \tilde{\mathbf{P}}_o^* \tilde{\mathbf{R}} \tilde{\mathbf{P}}_i^*$, which was then transformed back to the position-space matrix $\mathbf{R}_c$ (Fig. 2f). The retrieved $\phi_i(\mathbf{k}_i)$ and $\phi_o(\mathbf{k}_o)$ were almost identical due to optical reciprocity (Fig. 2g, h). The number of angular modes $N_c$ used for aberration correction in the pupil was about 6200, which is set by the FOD size of $40 \times 40$ μm². The magnitudes of the diagonal elements of $\mathbf{R}_c$ were greatly enhanced owing to the compensation of the sample-induced aberrations, and those of the off-diagonal elements were reduced in comparison to the original matrix $\mathbf{R}$ (Fig. 2c). Compared to the original OCM image (Fig. 2d), the intensity of the aberration-corrected image (Fig. 2i) acquired from the diagonal elements of $\mathbf{R}_c$ was increased by an average of about 30 times. The spatial resolution of the corrected image was estimated to be 450 nm, identical to the ideal diffraction limit.

We would like to emphasize that a new algorithm was added to the original CLASS algorithm in imaging through a mouse skull to correct local position-dependent aberrations ("Methods" and Supplementary Note 6). In complex scattering layer, the assumption that $P_i(\mathbf{r}; \mathbf{r}_i) = P_i(\mathbf{r} - \mathbf{r}_i)$ is valid only within the small isoplanatic patch. This means that aberrations cannot be corrected by a single pair of $\phi_i(\mathbf{k}_i)$ and $\phi_o(\mathbf{k}_o)$. We developed an algorithm to identify multiple $\phi_i(\mathbf{k}_i)$ and $\phi_o(\mathbf{k}_o)$ functions for

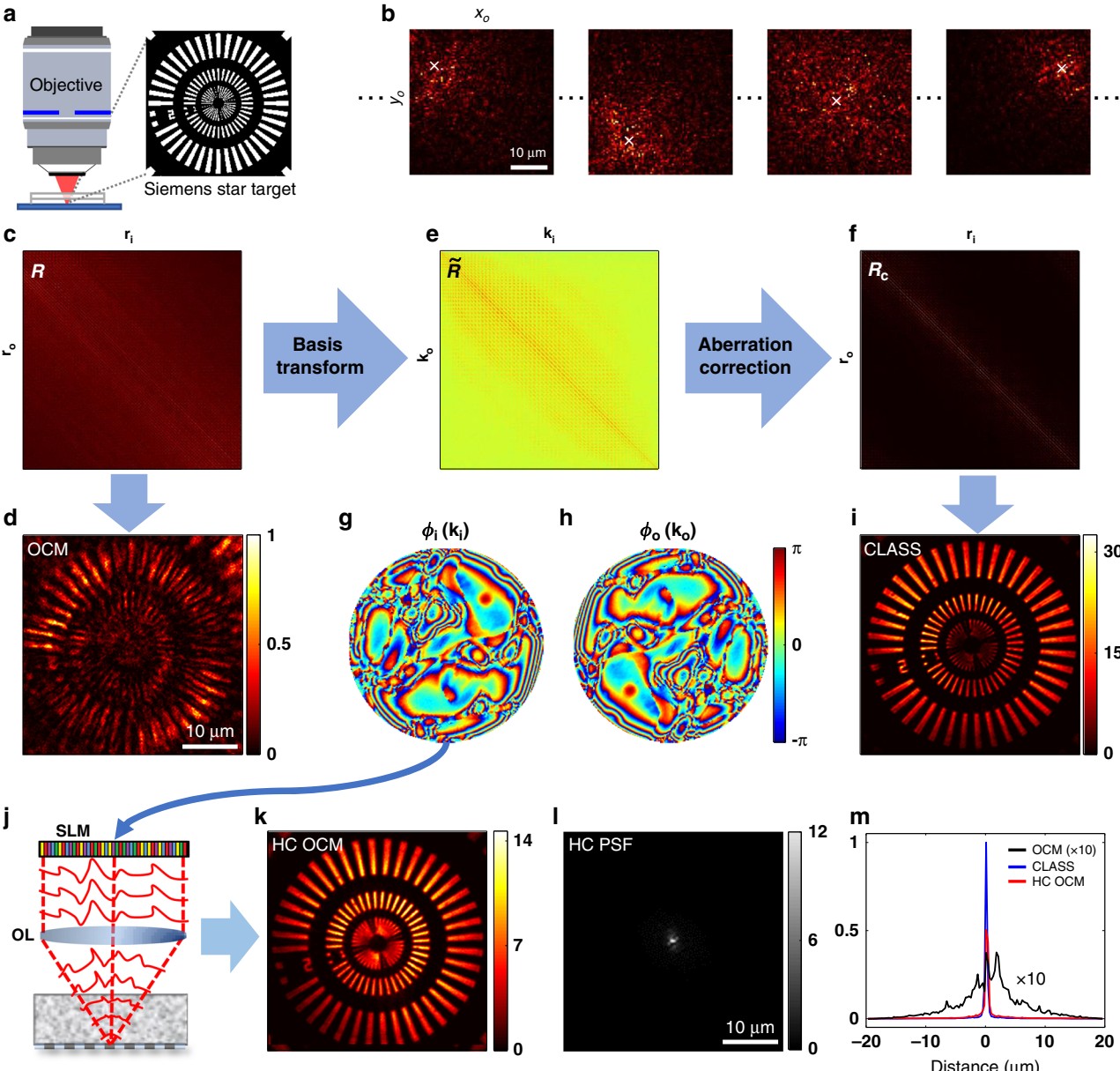

**Fig. 2 Aberration correction by the time-resolved reflection matrix. a** Sample geometry; a custom-made Siemens star target was placed under a 600-μm-thick, rough-surfaced plastic layer exhibiting strong aberrations. **b** Set of E-field images of reflected waves $E_{lab}(\mathbf{r}_o; \mathbf{r}_i)$ in the laboratory coordinates $\mathbf{r}_o = (x_o, y_o)$. Four representative amplitude images are shown. The white " ✕ " marks indicate illumination points $\mathbf{r}_i$. The FOD defined by the image size was set to $40 \times 40$ μm². Scale bar, 10 μm. **c** Time-resolved reflection matrix $\mathbf{R}(\mathbf{r}_o; \mathbf{r}_i)$ in the position space constructed from the set of E-field images in **b**. Each image was converted to a column vector and assigned to its corresponding column in $\mathbf{R}$. **d** OCM intensity image constructed from the main diagonal of $\mathbf{R}$ in **c** before aberration correction. Scale bar, 10 μm. **e** Reflection matrix $\tilde{\mathbf{R}}$ in spatial frequency space. **f** Aberration-corrected reflection matrix $\mathbf{R}_c$ converted from $\tilde{\mathbf{R}}$ after application of CLASS algorithm. **g, h** Phase maps $\phi_i(\mathbf{k}_i)$ and $\phi_o(\mathbf{k}_o)$ for aberrations in illumination and detection pupils retrieved by the CLASS algorithm, respectively. The radii of the maps correspond to a numerical aperture of 1.0. The number of modes $N_c$ used for aberration correction in the pupil was about 6200. **i** Aberration-corrected CLASS intensity image obtained from the main diagonal of $\mathbf{R}_c$ in **f**. **j** Schematic of hardware wavefront correction. Conjugate of the illumination pupil phase map in **g** displayed on the SLM in Fig. 1a to physically compensate for the aberrations. **k** OCM intensity image after hardware correction (HC) of aberrations. **l** Intensity image of reflected PSF measured at the camera after physical aberration correction by SLM (HC PSF). Scale bar, 10 μm. **m** Line profiles of the PSFs obtained without wavefront correction (black), after computational wavefront correction by the CLASS algorithm (blue), and after hardware wavefront correction by SLM (red).

individual isoplanatic patches. The algorithm is special in that the number of correction modes is maintained at the number of angular modes in full FOD (~10,000) even when the image analysis area is reduced to the isoplanatic patch size of $10 \times 10$ μm². This is contrary to conventional computational AO, where the number of correction modes is reduced in proportion to the image analysis area.

**Physical aberration correction by using SLM.** The LS-RMM system was designed not only to identify high-order aberrations induced by the medium but to physically compensate for them to generate a near-diffraction-limited focus on the sample. To this end, the SLM in the experimental setup (Fig. 1a) was used to display the phase conjugation of the identified input aberration map (Fig. 2g) to compensate for the wavefront distortions for

waves incident to and outgoing from the sample all together (Fig. 2j). With this hardware correction in place, we could obtain a clean OCM image (Fig. 2k) with no need for computational correction by the CLASS algorithm. The image quality was comparable to that seen in the CLASS image (Fig. 2i), which confirms that a sharp focus was physically generated at the plane of the target object. The PSF came into sharp focus, with a significantly increased peak intensity, after hardware wavefront correction by the SLM (Fig. 2l). We assessed the quality of the aberration correction by measuring the intensity profile across the center of the PSF (red curve, Fig. 2m). The original PSF of the reflected wave (black curve, Fig. 2m) was highly speckled with increased width. The enhancement by the Strehl ratio, measured by the ratio of the peak intensities after and before aberration correction, was 26, and the width of the focus approached the diffraction-limited spatial resolution of 450 nm. Peak height was about 50% of the computational aberration correction by the CLASS algorithm (blue curve, Fig. 2m), due to SLM limitations in shaping steeply varying aberrations, especially in the high spatial frequency range. The width of PSF, however, was almost identical to that of the computational correction.

**In vivo imaging of a mouse brain through an intact skull**. We tested the capacity for the CLASS algorithm to correct extreme, high-order aberrations, by attempting to image neuronal structures in the mouse brain through an intact skull. The mouse skull is composed of fine microstructures that induce severe optical aberrations as well as strong multiple-scattering noise. Thus far, only AO TPF fluorescence imaging[10,24,35] or three-photon excitation imaging[36] with excitation wavelengths greater than 1300 nm have been able to visualize mouse brain structures through the skull; label-free reflectance imaging has not yet been able to do so. This is because the round-trip aberrations jointly deteriorate the image quality, while input aberration matters most in multi-photon microscopy. The mouse skull consists of several layers of microstructures; these reduce the size of the isoplanatic patch down to the degree that confounds conventional AO methods[30]. In this study, we made significant improvements to the CLASS algorithm to correct aberrations locally with an isoplanatic patch size as small as $10 \times 10 \ \mu m^2$ while maintaining the number of correction modes to ~10,000 modes (Supplementary Note 6), ameliorating the extreme aberrations caused by the skull.

Figure 3 shows CLASS images of myelinated axons in the first cortex layer after different types of sample preparation. We first conducted imaging through a thinned skull (Fig. 3a–d), wherein compact bone and spongy bone were removed from the skull of a 7-week-old mouse to reduce its thickness to approximately 40 μm (Fig. 3a). We recorded a time-gated reflection matrix $R$ with a FOD of $30 \times 30 \ \mu m^2$ for the FOI of $150 \times 150 \ \mu m^2$ at a depth of 72 μm from the upper surface of the skull. In a conventional OCM image (Fig. 3b) reconstructed by the diagonal elements of $R$, the shapes of microvessels (blue arrowheads) and the myelinated axonal fibers (yellow arrowheads) appeared blurred. The effective diameter of a confocal pinhole used in the OCM image was set to 1 Airy unit. To correct anisoplanatic aberrations over a large field of view, we divided the entire view field into $18 \times 18$ regions and applied the CLASS algorithm to the individual subregions. Both microvessels and myelinated axons were visible with improved sharpness and signal strength in the aberration-corrected image (Fig. 3c). The retrieved aberrations varied spatially, but the degree of aberration was mild (Fig. 3d) due to the thinning of the skull. The size of each subregion was approximately $11 \times 11 \ \mu m^2$, including margins that overlapped with adjacent subregions. The number of correction modes for each aberration map, set by the FOD, was about 3500.

We then conducted imaging through an unaltered, intact skull excised from an 8-week-old mouse. In the 3D image reconstructed by SHG signals detected at the PMT, compact bone, spongy bone, and meninge could be discretely identified (Fig. 3e). Using the SHG image, the thickness of the skull was measured to be 90–110 μm. The time-gated reflection matrix $R$ was measured with a FOD of $50 \times 50 \ \mu m^2$ for the FOI of $150 \times 150 \ \mu m^2$ at a depth of 150 μm below the upper surface of the skull (red dotted box in Fig. 3e). OCM (Fig. 3f) and CLASS images (Fig. 3g) were obtained along with local aberrations (Fig. 3h). Compared to the thinned skull, the myelinated axons were virtually invisible in the OCM image because the thicker skull induced more complex aberrations. Our advanced CLASS algorithm was able to compensate for these increased aberrations and reconstruct myelinated axons at a high spatial resolution and contrast (Fig. 3g). The smallest thickness of the myelinated axons was measured to be 450 nm, confirming that the ideal diffraction-limited resolution was recovered. The aberration maps identified by our advanced CLASS algorithm with the same patch size as the thinned skull show increased complexity, and the correlation between aberration maps of adjacent regions was reduced. The number of correction modes in each map was about 10,000. From the obtained pupil aberration maps, it is estimated that the PSF width of the aberrated single-scattered waves is about 6–8 μm in full width at half maximum, which causes a reduction in the peak intensity of the single-scattering signal in the confocal spots by a factor of ~400. By comparing the PSFs before and after aberration correction, we estimate that the ratio of single-scattering signal to time-gated multiple-scattering background noise at confocal points was initially about 0.08, much smaller than 1, before the aberration correction. This explains why the conventional OCM failed to achieve high-resolution imaging of the mouse brain. The CLASS algorithm selectively refocused the aberrated single-scattering signals back to the confocal points to raise the single-scattering intensity by a factor of ~400. This made the single-scattering intensity larger than time-gated multiple-scattering noise by about 30 times after the aberration correction and enabled us to identify individual myelinated axons with diffraction-limited resolution (see Supplementary Note 7 for detailed PSF analysis).

Finally, we conducted in vivo imaging of myelinated axons through the intact skull of a living mouse (Fig. 3i–m). After removing the scalp from an 8-week-old mouse (Fig. 3i), a 100-μm-thick cover glass was glued on the surface of the intact mouse skull. A skull holder was attached to the mouse head with dental cement, which was then firmly fixed onto the custom-built stage. The thickness of the skull was measured to be ~120 μm. We measured the time-gated reflection matrix $R$ at a depth of 200 μm below the skull with the FOD of $30 \times 30 \ \mu m^2$ and obtained an OCM image from the diagonal elements (Fig. 3j). The aberrations were pronounced, such that no structures could be resolved in the OCM image. We then applied the CLASS algorithm to a $30 \times 30 \ \mu m^2$ sized area (blue dotted box, Fig. 3j) and obtained an aberration-corrected image (Fig. 3k). The reconstructed image shows signatures suggesting the existence of certain structures, but they could not be well resolved because the size of the FOI was larger than the isoplanatic patch size of the skull.

We addressed the local aberrations by dividing the area into $2 \times 2$ subregions and applied the CLASS algorithm to each subregion (Fig. 3l). Bright fiber structures began to emerge in the reconstructed image, and the retrieved aberration map in Fig. 3l also showed more complex phase structures. We further reduced the size of the subregion to $10 \times 10 \ \mu m^2$ and reconstructed the CLASS image (Fig. 3m). Myelinated axons were resolved without distortion following the aberration correction. Aberrations were complex, and the identified aberration maps resembled the

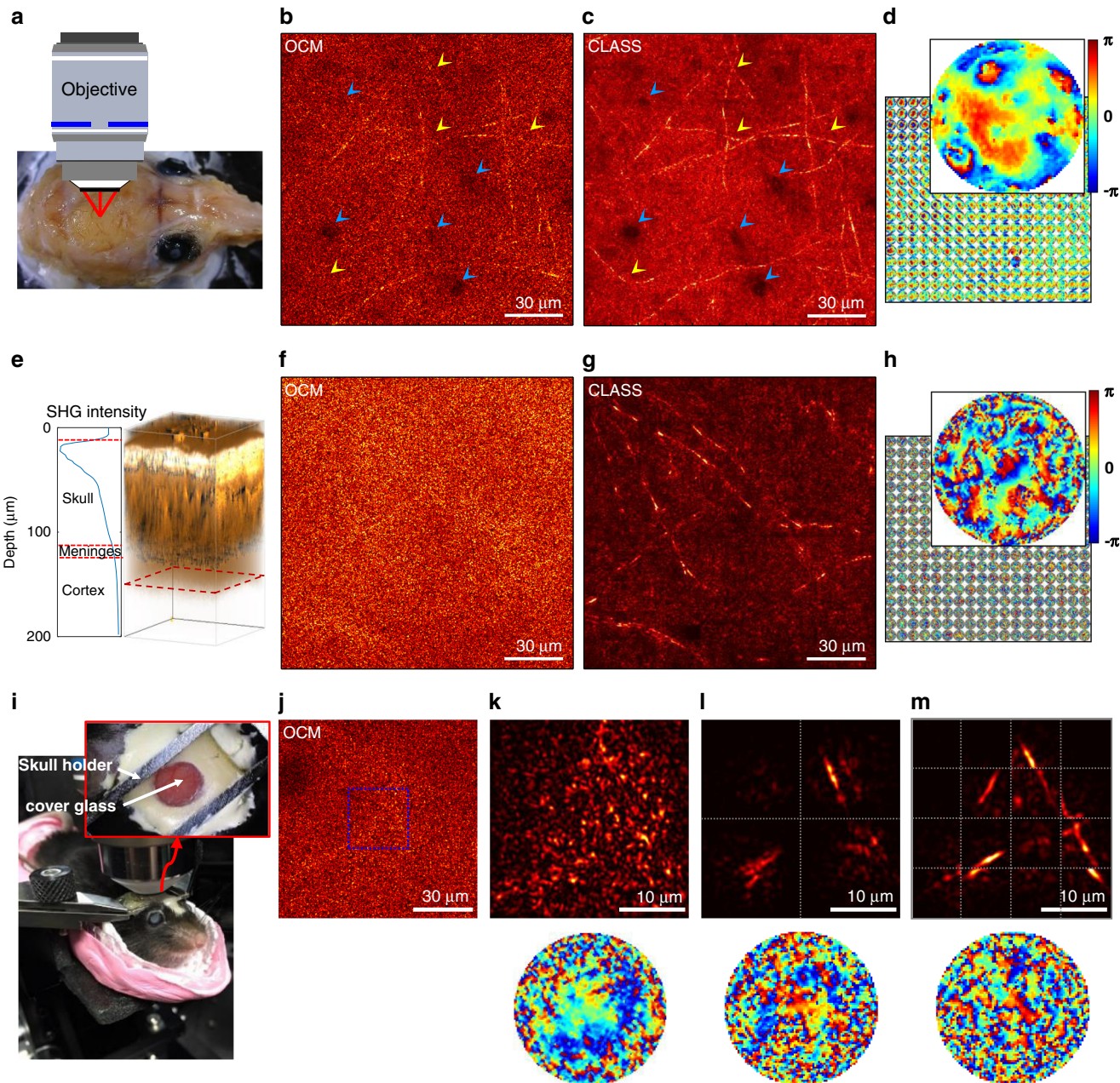

**Fig. 3 Imaging of myelinated axons through a mouse skull. a** Ex vivo specimen of a fixed mouse head. **b–d** Ex vivo imaging of mouse brain through the thinned skull of a 7-week-old mouse. The thickness of the thinned skull was approximately 40 μm. **b** Conventional OCM amplitude image of myelinated fibers in the first cortex layer at a depth of 70 μm from the upper side of the thinned skull. **c** Aberration-corrected CLASS amplitude image. The CLASS algorithm was individually applied to 18 × 18 subregions to correct local aberrations, and the corrected images were stitched together. The size of each subregion is ~11 × 11 μm², including overlap with adjacent areas. Microvessels (blue arrowheads) and myelinated axons (yellow arrowheads) are more clearly visible in the CLASS image. **d** Retrieved aberration maps for the detection pathway. The magnified map shows a representative aberration map. The number of correction modes $N_c$ for each aberration map in **d**, set by the FOD, was about 3500. Color bar, phase in radians. **e–h** Ex vivo imaging through intact skull of an 8-week-old mouse. The thickness of the skull was about 100 μm. **e** 3D reconstruction of SHG imaging of an intact mouse skull. The dashed red box indicates the depth at which CLASS imaging was performed. It was located 150 μm below the upper surface of the skull. **f** OCM amplitude image of myelinated fibers. Myelinated fibers are almost invisible. **g, h** CLASS amplitude image and corresponding aberration maps, respectively. The image was analyzed as in **c** and **d**. $N_c$ in **h** was about 10,000. **i** Experimental setup for in vivo imaging through intact mouse skull. The thickness of the skull was 125 −150 μm. **j** OCM intensity image of myelinated fibers at a 200-μm depth from the upper surface of the skull. **k** CLASS intensity image (upper) and the corresponding aberration map (lower) for a 30 × 30 μm² area marked by the blue dotted box in **j**. **l, m** CLASS intensity images with their representative aberration maps for one of the 2 × 2 and 4 × 4 subregions, respectively. $N_c$ used in aberration maps in **k–m** was about 3500. The skull thicknesses and imaging depths reported here are the raw axial movement of the sample stage.

speckle pattern. Of note, the application of the CLASS algorithm to a subregion size greater than $10 \times 10\ \mu m^2$ failed to resolve the fine structures of myelinated axonal fibers, suggesting that the isoplanatic size of the skull was about $10 \times 10\ \mu m^2$. When the subregion is larger than the isoplanatic patch, local aberrations are averaged in such a way that only the slowly varying aberrations in the pupil plane can be corrected; as such, the aberration map in Fig. 3k is much smoother than that in Fig. 3m. The isoplanatic patch size can systematically be found by finding the size of the subregion at which the intensity of the aberration-corrected image is maximum (see Supplementary Note 8). To the best of our knowledge, these results represent the first experimental demonstration of label-free reflectance imaging through an intact mouse skull at an ideal diffraction-limited resolution.

**Near-diffraction-limited TPF imaging through an intact mouse skull.** The LS-RMM can directly be combined with multiphoton microscopies such as TPF microscopy and second-harmonic generation (SHG) microscopy to recover their diffraction-limited spatial resolution in deep-tissue imaging. The experimental setup needs little modification, and all needed is to place a PMT right behind the dichroic mirror for detecting the nonlinear emissions from the specimens (Fig. 1). Here, we demonstrated near-diffraction-limited TPF imaging of neuron's dendrites through an intact skull. We prepared a fixed whole-mouse brain with an intact skull from a 4-week-old transgenic mouse that expresses enhanced green fluorescent proteins (EGFP) at the neuronal membranes (see "Methods" for sample preparation). We first performed CLASS imaging to obtain the aberrations induced by the mouse skull. The objective focus was set to $125\ \mu m$ below the upper surface of the skull, whose thickness was about $85\ \mu m$. Conventional OCM failed to visualize any myelin due to complex aberrations by the skull (Fig. 4a). We divided the entire field of view into $14 \times 14$ subregions and applied the CLASS algorithm for each subregion to recover fine myelination structures therein (Fig. 4b) and to obtain the aberration map at the corresponding subregion (Fig. 4c). The number of correction modes in each aberration map was 9880.

Similar to the demonstration in Fig. 2j–m, we were able to physically correct the skull-induced aberrations by displaying the phase conjugations of the aberration maps on the SLM. Since the aberration varied depending on the position, we made a physical correction for each subregion at a time. As a point of reference, a conventional TPF image was taken without aberration correction. Figure 4d shows a maximum intensity projection (MIP) of a volume image obtained in a depth range of 119–135 μm with 0.5-μm axial spacing over the same field of view like that in Fig. 4a. The contours of neuronal dendrites are blurry, and their microstructures are invisible due to the significant broadening and attenuation of the excitation PSF. For the subregion marked by the yellow dotted box in Fig. 4d, we applied the hardware correction using the aberration map obtained from the same subregion in Fig. 4b and conducted TPF imaging. The TPF image with hardware aberration correction (Fig. 4e) showed 19 times increased in fluorescence intensity and enabled the recovery of the fine dendritic spines that were invisible before the correction. Notably, only the structures in and around the subregion were corrected properly, supporting that the isoplanatic patch size is about the same as the subregion indicated by the yellow box, which is $10 \times 10\ \mu m^2$. For the full mapping of the neuronal dendrites, we conducted TPF imaging for each subregion indicated by the dotted gray box in Fig. 4b by displaying the phase conjugation of the associated aberration map in Fig. 4c. Multi-depth TPF images were obtained by scanning the imaging depth over the same depth range in Fig. 4d, where the imaging

depth was scanned by adding a defocus phase map to the SLM. Figure 4f, g show MIP images for a depth range of $113 \pm 1.5\ \mu m$ before and after aberration correction, respectively. Figure 4h, i are the same as Fig. 4f, g, respectively, but for a depth range of $122 \pm 1.5\ \mu m$. The corrected view field was expanded as expected, and dendrites and spines at two different depths were resolved. The measured width of the dendritic spine was as small as 500 nm, close to the diffraction limit.

TPF imaging that we demonstrated here can be considered a type of wavefront-sensing AO. LS-RMM serves as a tool to measure wavefront distortion by scattering tissues, similar to Shack–Hartmann wavefront sensors and other interferometric microscopy techniques. The uniqueness of our approach is that LS-RMM can identify extremely severe skull-induced aberrations without using either external guide stars or nonlinear fluorescence excitation. This benefit comes from the recording of the time-resolved reflection matrix and the use of the CLASS algorithm extracting the one-way aberration. In LS-RMM, the identification of wavefront aberrations is based on the intrinsic reflectance contrast. Myelinated axons are a good source of intrinsic contrast as they are spread throughout the mouse brain. It is not always necessary for the myelinated axons to be around the region of interest for multi-photon imaging. Even for the area where there are no axons such as the subregion indicated by the white arrowhead in Fig. 4b, we were able to identify the aberration map and obtain near-diffraction-limited TPF images. This is because noise-like speckle patterns backscattered from irregular tissue structures at the focal plane are also single-scattered waves contributing to the CLASS algorithm.

Adaptive optics by LS-RMM is beneficial over existing AO modalities that rely upon feedback optimization of multi-photon fluorescence image in that photobleaching effect is negligible during the measurements of the aberrations as it does not require high excitation power to record the reflection matrix. We used an excitation power of 21 mW at the sample and a pixel exposure time of 100 μs for TPF imaging, whereas an excitation power of 3 mW was used for recording the reflection matrix. Another important benefit is that recording of one reflection matrix over the area of $150 \times 150\ \mu m^2$ enabled us to obtain a $15 \times 15$ number of aberration maps for as many different isoplanatic patches. In the iterative feedback AO, multiple iterations of optimization need to be conducted for each isoplanatic patch.

## Discussion

We developed a laser-scanning reflection-matrix microscopy (LS-RMM) and streamlined the closed-loop accumulation of single-scattering (CLASS) algorithm for correcting local high-order aberrations even in the presence of strong multiple-scattering noise. The system is highly compatible with the conventional laser-scanning confocal microscopy as it shares the same backbone, but it is distinct from conventional confocal imaging in that backscattering signals arriving at both non-confocal and confocal positions are detected. We constructed a time-gated reflection matrix in the space domain and corrected sample-induced wavefront aberrations without requiring guide stars. In particular, we improved the aberration correction algorithm to correct up to 10,000 angular modes of aberrations locally varying at every 10-μm interval in the sample plane. We thus demonstrated label-free reflectance imaging of cortical myelination with extremely complex local aberrations caused by the intact skull of a living mouse. Since the aberration correction takes place computationally in the coherent reflectance imaging, the recording of the reflection matrix itself is the end of the imaging session. This is particularly beneficial for in vivo imaging because we have ample time to correct severe position-dependent aberrations after the

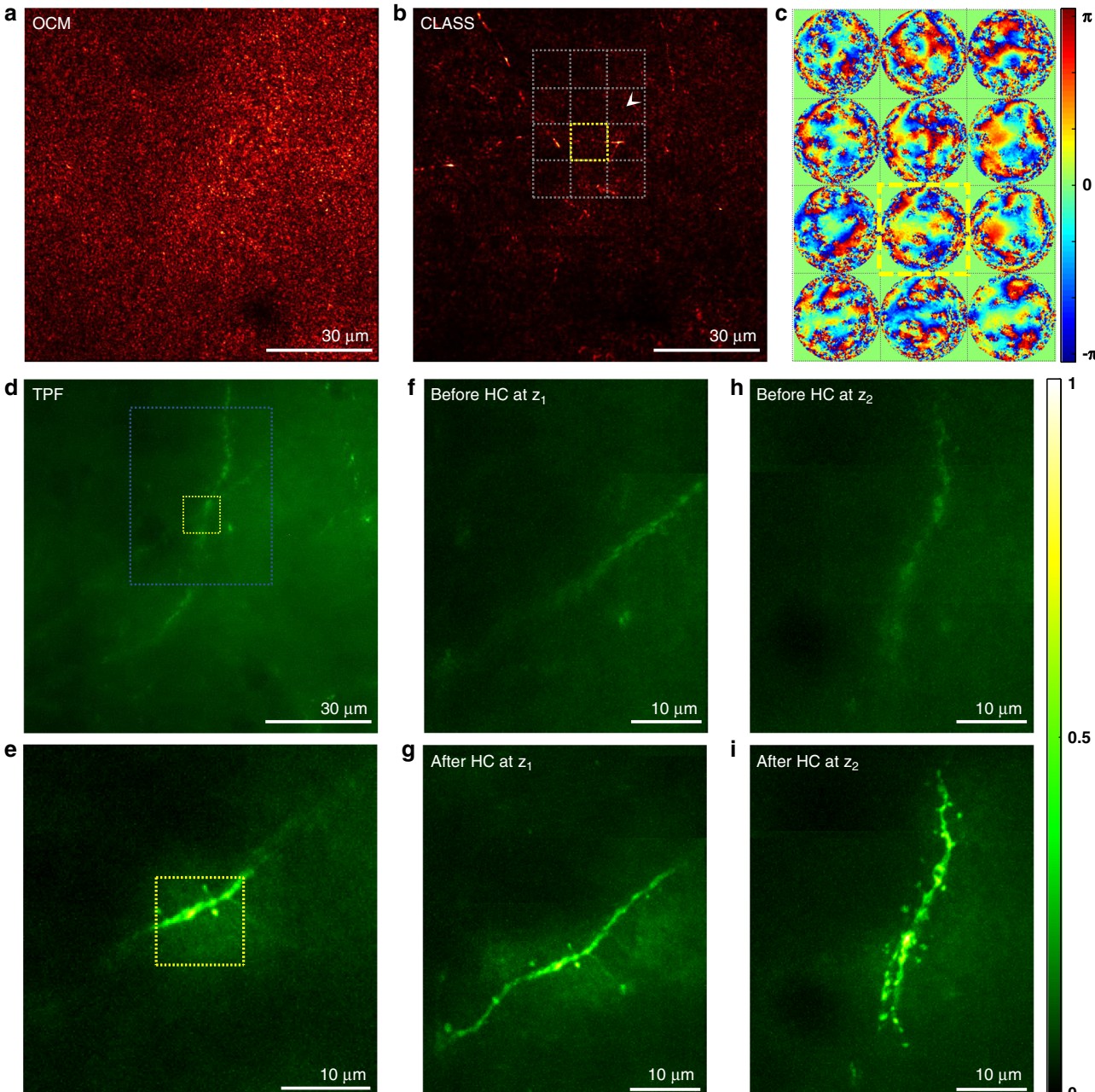

**Fig. 4 TPF imaging through an intact mouse skull. a** Conventional OCM image under an intact mouse skull before aberration correction. The thickness of the skull was about 85 μm, and the focal plane was set to a depth $z_0 = 125$ μm from the upper surface of the skull. **b** MIP of five LS-RMM images taken over a depth range of 117–133 μm with 4-μm steps. LS-RMM image for each depth was obtained by stitching aberration-corrected 15 × 15 subregions. Note that myelinated fibers appear discontinuous in the image mainly due to coarse depth steps and large inclination angles of the fibers with respect to the image plane. **c** Aberration maps of subregions at $z_0 = 125$ μm indicated by the gray dotted boxes in **b**. The size of the subregion is 10 × 10 μm², and each phase map contains 9880 angular modes. Color bar, phase in radians. **d** MIP of TPF images at the same position as **a** before hardware aberration correction. The MIP image was obtained for a depth range of 119–135 μm with a 1.5-μm increment. **e** TPF image after physical aberration correction for the subregion indicated by the yellow box in **b**. Yellow boxes in **d** and **e** correspond to the same yellow box area in **b**. **f, g** MIP of TPF images at the depth $z_1 = 113 \pm 1.5$ μm before and after aberration correction, respectively, for the area, indicated as a white dashed box in **d**. **h, i** Same as **f** and **g**, respectively, for the depth $z_2 = 122 \pm 1.5$ μm. Color bar, intensity normalized by the maximum intensity in **i**. Scale bars indicate 30 μm in **a**, **b** and **d**, and 10 μm in **e–i**.

data acquisition. Still, the data acquisition speed is critical for in vivo imaging because even small perturbation for the reflection matrix can undermine the image reconstruction, especially in the case of severe aberrations. Reflectance imaging is more challenging than fluorescence imaging in that the contrast of the reflectance imaging of living tissues is intrinsically low. Furthermore, the round-trip aberrations jointly deteriorate the point-spread-function while only the aberration in the excitation beam matters most in the fluorescence imaging. Our LS-RMM addressed all these difficulties for us to realize the first through-skull reflectance imaging to date.

The working depth of current implementation is set by the weak reflectance contrast of myelinated axons relative to the multiple-scattering noise. LS-RMM can employ any excitation

wavelength, similar to confocal reflectance imaging. Therefore, the use of longer excitation wavelengths can help to increase imaging depth as the multiple-scattering noise is to be reduced[37]. Alternatively, the use of visible wavelengths will lead to increasing the image contrast and spatial resolution, which will open the possibility of investigating detailed myelin pathologies through intact skull[38]. Regarding the aberration correction algorithm, previous reflection-matrix approaches often use singular value decomposition to cope with multiple-scattering noise and/or aberrations[31,34,39,40]. In the context of optical imaging, it was demonstrated to the extent that a sharp image was recovered for a few-micron-sized highly reflecting resolution target hidden under a scattering and aberrating tissue[34]. On the contrary, our proposed algorithm works faithfully for the recovery of diffraction-limited resolution of weakly reflecting biological specimens embedded within biological tissues under a rat skull.

In addition to the reflectance imaging, we physically corrected the skull-induced aberrations by using an SLM in the excitation beam path and generated a sharp optical focus within brain tissues under the skull. This physical aberration correction was applied to the through-skull multi-photon imaging. We demonstrated TPF imaging of neuronal dendrites and their minute spines with the spatial resolution of 500 nm, close to the diffraction-limited resolution of 380 nm. Notably, multiple aberration maps were displayed in sequence for individual $10 \times 10$ $\mu m^2$ subregions to recover dendrites over the field of view much larger than the isoplanatic patch size set by the skull. Furthermore, we also demonstrated the near-diffraction-limited SHG imaging of collagen fibers underneath an intact mouse skull (Supplementary Note 9). In LS-RMM, the identification of wavefront aberrations is based on the intrinsic reflectance contrast of targets. As such, it does not require fluorescent labeling and high excitation power, contrary to existing AO modalities that rely upon multi-photon fluorescence feedback signals. Another critical advantage of LS-RMM is that the computational process to retrieve both the wavefront aberrations and the aberration-corrected reflectance image takes place after the recording of the reflection matrix. The hardware aberration correction capability of LS-RMM can also be added to other imaging modalities such as super-resolution imaging[41,42] and coherent Raman imaging[43,44] to enhance their imaging depth in deep-tissue imaging. The practical limitation of our approach for in vivo multi-photon imaging is the computation time for the CLASS algorithm. It does not take long to process the algorithm for mild aberrations from typical brain tissues. However, it is time-consuming to process the skull-induced aberrations with an extremely small isoplanatic patch, which could be improved through the use of graphics processing units (GPUs) or cluster-based hardware; algorithm improvements in the future could also shorten this step.

## Methods

### Experimental setup in Fig. 1a.
A mode-locked Ti:Sapphire laser (center wavelength, 900 nm; bandwidth, 25 nm; repetition rate, 80 MHz) was used as a short-coherence-length light source (coherence length, 30 μm). The laser beam was split into sample and reference beams by a beam splitter (BS1), and they were recombined by another beam splitter (BS2) to form Mach–Zehnder interferometry. The sample beam was relayed via two GMs and focused on the sample plane by a water immersion objective lens (OL, Nikon, ×60, NA 1.0). The GMs were used to raster-scan the focused illumination beam on the sample as in a conventional confocal microscope. The backscattered wave from the sample was collected by the same OL and traveled through a reciprocal path. The wave was then de-scanned by the GMs and delivered to a camera (s-CMOS, pco.edge 4.2, PCO AG) placed at the conjugate image plane. The reference beam was sent through multiple pairs of relay lenses, an optical delay line (not shown), and a diffraction grating (DG). The first-order diffraction of the reference beam by the DG was selected for by an iris diaphragm (I) and delivered to the camera as a plane wave. Consequently, its phase front was tilted with respect to the camera plane while its temporal pulse front remained parallel to the camera plane. The interference between the sample and

reference waves formed an interferogram within the temporal gating width corresponding to 15 μm in length, half the coherence length of the light source.

### The acquisition time for the full reflection matrix.
The frame rate $f_{cam}$ of the s-CMOS camera used here is given by $f_{cam} = r/\sqrt{FOD}$ [Hz], wherein the experiment, the coefficient $r$ was 40,000 [1/μm]. The number of correction modes $N_c$ in a circular pupil with a numerical aperture, $\alpha$ is given by $N_c = \pi(\sqrt{FOD}/2\delta_d)^2$, where $\delta_d = \lambda/2\alpha$ is the diffraction-limited resolution. For example, an FOD size of $50 \times 50$ $\mu m^2$ covers approximately 10,000 correction modes for $\alpha = 1$ and $\lambda = 900$ nm. The number of images to be recorded for complete measurement of a reflection matrix for a given FOI is derived by $N_s = FOI/\delta_d^2$. The total acquisition time for a full reflection matrix is then determined by $T = N_s/f_{cam}$. For the FOI of $50 \times 50$ $\mu m^2$ and FOD of $50 \times 50$ $\mu m^2$, it takes about 15 s to measure the reflection matrix. At a smaller FOD size of $16 \times 16$ $\mu m^2$ containing about 1000 correction modes, the data acquisition time is reduced to around 5 s. The use of a high-speed camera along with optimization of the algorithm could potentially reduce this acquisition time to below 1 s.

### Aberration correction using the CLASS algorithm.
The CLASS algorithm is an iterative method for finding the aberration maps $\phi_i(k_i)$ and $\phi_o(k_o)$ independently from $\bar{R}(k_o; k_i)$. In $n$th iterative step, the trial solutions for the aberration maps, $\phi_i^{(n)}(k_i)$ and $\phi_o^{(n)}(k_o)$, and the aberration-corrected reflection matrix, $\bar{R}^{(n)}(k_o; k_i)$, are obtained by the following iteration formulas,

$$\phi_i^{(n)}(k_i) = \arg\left\{\sum_{\Delta k} \bar{R}^{(n-1)}(\Delta k; k_i)^* \cdot \bar{R}_{CLASS}^{(n-1)}(\Delta k)\right\}, \quad (4)$$

$$\phi_o^{(n)}(k_o) = \arg\left\{\sum_{\Delta k} \bar{R}^{(n-1)}(k_o; \Delta k)^* \cdot \bar{R}_{CLASS}^{(n-1)}(-\Delta k)\right\}, \quad (5)$$

$$\bar{R}^{(n)}(k_o; k_i) = \exp[i\phi_o^{(n)}(k_o)] \cdot \exp[i\phi_i^{(n)}(k_i)] \cdot \bar{R}^{(n-1)}(k_o; k_i), \quad (6)$$

$$\bar{R}_{CLASS}^{(n)}(\Delta k) = \sum_{k_i} \bar{R}^{(n)}(\Delta k; k_i). \quad (7)$$

Here, $\Delta k = k_o - k_i$ is the momentum difference, and $\bar{R}^{(n)}(\Delta k; k_i)$ and $\bar{R}^{(n-1)}(k_o; \Delta k)^*$ are the elements of $\bar{R}^{(n)}(\Delta k; k_i)$ and $\bar{R}^{(n)}(k_o; \Delta k)$, respectively. The symbol * denotes the complex conjugate. Starting with an initial guess of $\phi_i^{(0)}(k_i) = \phi_o^{(0)}(k_o) = 0$ and $\bar{R}^{(0)} = \bar{R}$, the iteration is continued until the root-mean-square error of the $n$th aberration map becomes less than a predefined value. The final input and output aberration maps are then found by accumulating all the preceding aberration maps; $\phi_i(k_i) = \sum_n \phi_i^{(n)}(k_i)$ and $\phi_o(k_o) = \sum_n \phi_o^{(n)}(k_o)$.

### Advanced algorithm for correcting spatially varying high-order aberrations.
To correct local aberrations, it is necessary to analyze the FOI equal to or smaller than the isoplanatic patch given by the scattering medium. In conventional computational AO and the previous CLASS algorithm, dividing the entire FOI into small subregions accompanies the simultaneous reduction of FOI and FOD in each subregion. This results in the reduced $N_c$, making it difficult to correct complex aberrations in small isoplanatic patches. In this study, we developed an algorithm that can find local aberrations while $N_c$ is maintained to be large enough. In LS-RMM, we can set FOD independently of FOI to make use of the information outside the subregion. For example, in our in vivo imaging through the intact mouse skull (Fig. 3m), we set the FOD ($30 \times 30$ $\mu m^2$) larger than FOI ($10 \times 10$ $\mu m^2$). Consequently, the reflection matrix $R_{sub}(r_o; r_i)$ for each subregion became an $N_o \times N_i$ rectangular matrix, where $N_o$ is set by the number of pixels in $(\sqrt{FOD} + \sqrt{FOI})^2$ and $N_i$ is set by the number of pixels in FOI. Therefore, this local reflection matrix contains only 390 input modes set by FOI, but it contains $N_c = 3500$ output modes set by the FOD. In our advanced CLASS algorithm, we added zero columns to $R_{sub}(r_o; r_i)$ to convert it into an $N_o \times N_o$ square matrix. We then transformed it into $\bar{R}_{sub}(k_o; k_i)$ having the same grid spacing $\delta k = 2\pi/(\sqrt{FOD} + \sqrt{FOI})$ for both the input and output pupil planes. In the $n$th iteration, the output aberration map $\phi_o^{(n)}(k_o)$ was identified from Eq. (5) for all the $N_c$ modes supported by the FOD. Based on the optical reciprocity, we applied $\phi_o^{(n)}(-k_o)$ to $\phi_i^{(n)}(k_i)$ to find $\bar{R}_{sub}(k_o; k_i)$ in Eq. (6). By repeating this process, we were able to correct both the input and output aberrations for all the $N_c$ angular modes even though the FOI is reduced to the size as small as an isoplanatic patch. Due to this capacity, we could identify locally varying high-order aberrations (Fig. 3m). For detailed analysis, see Supplementary Note 6.

### Sample preparation for ex vivo imaging.
All animal experiments were approved by the Korea University Institutional Animal Care & Use Committee (KUIACUC-2019-0024). Three- to eight-week-old C57BL/6 mice or Thy1-EGFP line M (Jackson Labs #007788) mice were deeply anesthetized with an intraperitoneal injection of ketamine and xylazine at 100 and 10 mg/kg, respectively, and

decapitated. The scalps were completely removed, and the heads were fixed with 4% paraformaldehyde at 4 °C for 1–2 days. Heads were then washed with phosphate-buffered saline (PBS) in triplicate. For imaging through the intact skull, an unaltered head was affixed to a 35-mm dish using cyanoacrylate and filled with PBS. For imaging through the thinned skull window, compact and spongy bones were ground from a fixed head. The 3–4-millimeter area in the center of the parietal bone was thinned using a 0.5-mm surgical carbide bur in a high-speed dental drill. Afterward, the bone was smoothed with a stone bur and a polishing bur to a thickness of 30−40 μm.

**Preparation of intact skull window for in vivo imaging.** We followed the procedure outlined by Wang et al.[36] with slight modifications. Briefly, mice (6–8 weeks old, 22−28 g) were temperature-controlled and anesthetized with isoflurane (1.5 −2% in oxygen to maintain a breathing frequency of around 1 Hz). Dexamethasone (1 mg/kg) was administrated intramuscularly on the 2 consecutive days after the surgery to minimize swelling at the surgery site. Eyes were covered with eye ointment during surgery and imaging. The hair was removed with scissors and Nair hair remover. The scalp was removed to expose the bregma, lambda, and parietal plates. Sterile saline was applied to the skull, and connective tissue remaining on the skull was gently removed. A sterile, round, 4 mm (diameter) coverslip (#1, Warner Instruments, USA) was attached to the center of the parietal bone using ultraviolet-curable glue (Loctite 4305). A custom-made metal plate was attached to the skull with cyanoacrylate for head fixation, and the exposed part of the skull was covered with dental cement (Dentsply DeTrey GmbH, Germany).

**Reporting summary.** Further information on research design is available in the Nature Research Reporting Summary linked to this article.

## Data availability
The data that support the findings of this study are available from the authors on reasonable request.

## Code availability
The code used in this study is available from the corresponding author upon reasonable request.

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

## Acknowledgements
This work was supported by the Institute for Basic Science (IBS-R023-D1).

## Author contributions

W.C., S.Y., and H.L. conceived the project. S.Y. and H.L. designed the experimental setup, performed the experiments with the help of J.H.H., and analyzed data. J.H.H. prepared biological samples. Y.S.L. supported ultra-short pulse laser system. S.Y., H.L., and W.C. prepared the manuscript, and all authors contributed to finalizing the manuscript. W.C. supervised the project. S.Y. and H.L. contributed equally to this work.

## Competing interests

The authors declare no competing interests.
