## [Peer Review File · Nature Communications]

Reviewers' comments:

Reviewer #1 (Remarks to the Author):

The manuscript presents the reflection matrix based aberration measurement and compensation method for imaging through media with scattering and aberration.

Different from the earlier implementation by the same team, the incident light is a focused beam (not a single direction, k vector beam). Compared to low coherence confocal detection (optical coherence microscopy) which only contains the information of one reflected spatial mode, the proposed method can record more information (no confocal detection, low coherence interferometry for nearby spatial modes) about the scattering media. Through computation, the method can determine the wavefront distortion encountered by the incident beam and the reflected beam.

The method can be used for not only computation AO compensation but also hardware AO compensation. To test the effectiveness of the AO measurement, they utilized hardware AO to best both OCM and SHG imaging through mouse skull, and OCM imaging of test target through rough surface glass layer.

In comparison, the method seems to work well for the test target imaging through rough plastic surface (aberration on surface, low scattering). However, for the through skull imaging (aberration and scattering), the performance only shows moderate improvement.

For the case of thinned skull, it seems that the only difference is in the signal strength (moderate increase). If the regular OCM imaging and SHG imaging were done with higher power, it may yield comparable image quality.

For the case of through intact skull imaging, the imaging results were not consistent. If we compare Fig. 3L and 3M, some of the features that are available in L were not visible in M while some features of M were not visible in L. Moreover, there is no ground truth for this measurement. I assume one can remove the skull and image the same area and provide a ground truth image of the axon.

In addition, there is not SHG imaging for the through intact skull imaging measurement. If the correction is effective, it should at least show large signal increases (second order signal).

Overall, the data seems to suggest that this method works better for aberration (not very scattering) sample but seems to have trouble working with scattering sample. The quality of the through-skull imaging is not great. It seems that regular OCM or SHG with a bit higher power can achieve the same image quality for the case of thinned skull imaging. The through skull imaging has no ground truth and no SHG image comparison. So it is hard to tell if the method indeed yields correct images.

I'd suggest that 1) provide ground truth image for comparison. Otherwise, we can not tell if the images were correct for the case of through skull imaging; 2) provide SHG image comparison for through intact skull data. A large signal increase in SHG can demonstrate the effectiveness of the correction. 3) more and better in vivo test results. Is the system really reliable and working well in most cases (so that people can use it consistently)? 4) discussion for the performance difference between the test target imaging (through rough plastic surface) and the through skull imaging (real biological scattering tissue).

Reviewer #2 (Remarks to the Author):

Dear Editor

In this work, the authors investigate a reflection matrix approach of label-free optical imaging that allows to overcome high-order spatially-varying sample-induced aberrations. In particular, the authors demonstrate diffraction-limited imaging of myelinated axons through ex-vivo mouse skulls. Compared to conventional adaptive optics, a reflection matrix approach does not require any guide star and allows to tackle space variant aberrations. Once the reflection matrix is recorded, the CLASS algorithm, developed in previous works by the same authors and improved in that paper, allows to correct in post-processing high-order aberrations both at input and output. The main novelty in the current paper is the measurement of the reflection matrix in a focused basis both at input and output, while, in previous papers, the authors were measuring the reflection matrix in a plane wave basis. Albeit more challenging due to shot-to-shot phase fluctuations, the focused basis ensures, according to the authors, "the optimal use of the detector's dynamic range, resulting in enhanced sensitivity to multiple scattering noise". This is an aspect with which I may disagree and that I will discuss below.

In overall, this paper is of excellent quality, well written, concise and clear. This work is useful both for researchers working in the field of wave-front shaping and those more generally motivated by the difficult and long-standing challenge of imaging in/through complex media. I think it perfectly fits with the scope of Nature Communications. For all these reasons, I would suggest to accept this paper. However, one main objection that can be addressed to the authors is the question of novelty of this work with regards to their own previous papers also published in Nature Communications [27,28]. I would suggest the authors to be clearer about the merit of this current work compared to previous ones. Otherwise, one may think this work is only incremental.

1/ With respect to this novelty issue, I am questioning the interest of recording the reflection matrix in a focused basis both at emission and reception rather than in a plane wave basis. I am not sure to understand why this basis allows the optimal use of the detector's dynamic range. In a focused basis, most of the reflected wave-field is localized in the vicinity of the input focusing point while it spreads over the whole detector surface when the latter one is placed in the pupil plane. If we have a good photon budget, this last configuration seems to be better in terms of sensitivity since more photons can be detected before reaching the saturation of the camera. Could the authors comment on that?

2/ A second advantage of the focused basis invoked by the authors is the fact that "signals" can be more easily discriminated from the multiple scattering noise than in a plane wave basis. Indeed, the single scattering contribution is more likely to be focused in the vicinity of the input focusing point, while the multiple scattering noise spreads over the whole image plane. I agree but one could have recorded the reflection matrix in the pupil plane, project the data in the focused basis via a numerical Fourier transform (as the authors do) and apply an adaptive confocal filter on the data in the focused basis in order to restrict their field of illumination and/or detection as needed (as done in Ref.32). The reflection matrix in a plane wave or focused basis carries the same information. One can easily project the wave-field from one basis to another, depending on what we want to do (aberration correction, multiple scattering filter, etc.). Hence there does not seem to have much interest of recording the reflection matrix in a focused basis rather than in a plane wave basis, like the authors did in their previous works.

3/ The main improvement on the CLASS algorithm proposed in that paper is the possibility of considering a field-of-detection (FOD) much larger than the field-of-illumination (FOI). Indeed, for a local correction of position-dependent aberrations, the FOI should be restricted to an isoplanatic patch while the FOD should capture the whole imaging PSF. The latter parameter governs the number of angular modes over which aberrations can be corrected. While the authors discuss in depth the choice of the FOD as a compromise to be found on the signal-to-multiple scattering ratio, the choice of the FOI is not discussed at all. How do they a priori determine the size of an isoplanatic patch? How do they choose the sub-division of the field-of-view?

Besides these main points, I have few minor comments:

4/ The authors mention that they can remove most of the multiple scattering background by carefully choosing the FOD but they do not provide any quantification of the multiple scattering level in their experiments. Related to that question, do they have an estimation of the scattering or transport mean free paths in the mouse skull and brain such that they can express the imaging depth in terms of scattering/transport mean free path?

5/ Line 196: The authors say "The sample-induced aberrations are dominated by angle-dependent phase retardation". Why? Is it because the skull is far away from the focal plane and can thus be considered as a far-field phase screen?

6/ In relation to the previous point, is the CLASS algorithm limited angle-dependent phase retardation? In other words, can it also address space-dependent phase retardation effects induced by sample inhomogeneities close to the focal plane?

7/ Line 198: Why does the projection of the reflection matrix in the plane wave basis writes: $\tilde{R} = F^{-1} R F$? According to me, the Fourier transform should have the same convention at the input and output: $\tilde{R} = F^T R F$ (where the symbol T stands for transpose). Otherwise, the reflection matrix is no longer symmetrical while it should be by virtue of spatial reciprocity.

8/ Line 353: Title of Figure 3 -> I would say "through a mouse skull" rather than "in a mouse skull".

Reviewer #3 (Remarks to the Author):

Yoon et al. describe an imaging method that is essentially a form of aberration-corrected OCT microscopy, where the aberration correction is obtained from a diversity of illumination conditions. They combine digital and hardware aberration corrections. The power of their method is demonstrated by in vivo imaging of brain tissue through an intact mouse skull. This work shows great promise for various kinds of deep-tissue imaging and the experimental work is very high quality.

In general the manuscript is well written, however, there seems to be hedging of acronyms. The authors name their current method LS-RMM, however in the results section, the relevant images are indicated as CLASS, which refers to the data processing algorithm.

LS-RMM, as far as I understand from the manuscript text is a combination of a specific data collection strategy with an enhanced version of the CLASS algorithm. This is more or less clear by going back to the flow diagram in Fig. 2 but I would prefer it to be clearer from the text also.

The gain of the new method with respect to the previously published CLASS method is significant, but this is only made clear in the supplementary.

An important point is the relation to recent work from the Boccara group. A current preprint from that group <https://arxiv.org/abs/1910.07252> introduces quite similar ideas and utilizes essentially the same optical setup to measure through much thicker tissue (but employing brighter objects). The relation to this later work should be discussed, not only the older published work from the same group.

As a very minor point, the manuscript text needs some proofreading for grammar, especially placement of the articles "the" and "a".

The final proof by using the recovered aberration map for 2-photon microscopy is strong evidence that the method recovers the true object. Again, it is a pity that this is hidden in the

supplementary information, but probably length limitations do not allow otherwise.

In conclusion, this manuscript presents an aberration-corrected OCT method with the potential to image realistic objects through strongly aberrating thin media. I recommend it for publication with attention to the above points.

Author's Response to Reviewer #1:

The manuscript presents the reflection matrix based aberration measurement and compensation method for imaging through media with scattering and aberration.

Different from the earlier implementation by the same team, the incident light is a focused beam (not a single direction, k vector beam). Compared to low coherence confocal detection (optical coherence microscopy) which only contains the information of one reflected spatial mode, the proposed method can record more information (no confocal detection, low coherence interferometry for nearby spatial modes) about the scattering media. Through computation, the method can determine the wavefront distortion encountered by the incident beam and the reflected beam.

The method can be used for not only computation AO compensation but also hardware AO compensation. To test the effectiveness of the AO measurement, they utilized hardware AO to test both OCM and SHG imaging through mouse skull, and OCM imaging of test target through rough surface glass layer.

We appreciate the reviewer's taking the time to thoroughly review our manuscript and providing us with critical comments and suggestions. We addressed all of them carefully in this response letter and revised our manuscript accordingly. In particular, we performed new experiments applying LS-RMM for correcting the sample-induced aberrations in two-photon fluorescence (TPF) imaging. We demonstrated near-diffraction-limited TPF imaging of dendritic spines under an intact mouse skull, which is added to Fig. 4 of the revised manuscript. We think that the addition of this new data addresses most of the reviewer's major concerns.

In comparison, the method seems to work well for the test target imaging through rough plastic surface (aberration on surface, low scattering). However, for the through skull imaging (aberration and scattering), the performance only shows moderate improvement.

The reviewer seems to think that the performance of our method is only moderate mainly based on the fact that the image quality of our brain imaging is not as good as that of the test target. We would like to emphasize that the reconstructed images in Figs. 3c, 3g, and 3m show similar quality as the OCM image taken near the surface of bare brain tissue with no skull on the top (Fig. R1).

Figure R1. Conventional OCM image from mouse brain without skull. OCM image of cortex layer1 of brain tissue extracted from a 4-weeks-old mouse.

Even without aberration and multiple scattering noise, the contrast of label-free reflectance image of myelinated axons is quite low at the excitation wavelength of 900 nm. In the reflectance imaging, the source of contrast originates from the difference in reflectance between the lipid-rich myelinated axons and the surrounding tissues in the focal plane. Considering that the refractive index of the lipid bilayer is 1.42-1.45 and the average refractive index of brain tissues is 1.36-1.4, we are visualizing the reflectance difference by the refractive index contrast of 0.02-0.1. On the contrary, the test target has high reflectance contrast of near 100 %, and this is why the performance appears to be better. In fact, fluorescence imaging has similar benefit as the test target in terms of image contrast. Indeed, this weak contrast makes the label-free reflectance imaging extremely difficult in deep-tissue imaging and serves as one of the main reasons why reflectance imaging hasn't been realized through an intact skull so far. Once again, the contrast of the reconstructed image is lower than that of the resolution target due to the intrinsically low reflectance contrast of biological tissues, not due to the performance of our method.

We added the following sentences to the discussion section to signify the differences of the label-free reflectance imaging from the fluorescence imaging.

“Reflectance imaging is more challenging than fluorescence imaging in that the contrast of the reflectance imaging of living tissues is intrinsically low. Furthermore, the round-trip aberrations jointly deteriorate the point-spread-function while only the aberration in the excitation beam matters most in the fluorescence imaging. Our LS-RMM addressed all these difficulties for us to realize the first through-skull reflectance imaging to date.”

For the case of thinned skull, it seems that the only difference is in the signal strength (moderate increase). If the regular OCM imaging and SHG imaging were done with higher power, it may yield comparable image quality.

In the 50- μm -thick thinned skull imaging (Fig. 3c), the improvement in image resolution was rather moderate mainly because the aberrations were initially weak. Still, there are structures such as those indicated as yellow arrowheads in Fig. 3c that became visible only after the aberration correction. This occurred in areas where the aberrations were locally more severe. The improvement in resolving power was dramatic in the intact skull imaging as shown in Figs. 3f and 3g because the aberrations were much more complex than the thinned skull imaging. Conventional OCM almost completely lost resolving power for the intact skull imaging. The myelinated axons were invisible throughout the entire view field before the aberration correction with the diffraction-limited spatial resolution.

While we corrected complex aberrations shown in Fig. 3h, which correspond to Strehl ratio enhancement of 400 times, the increase in the image intensity wasn't high as much. In fact, the image intensity at the myelinated axons was enhanced only by 30 times. The main reason is due to the time-gated multiple scattering noise. Initially, multiple scattering noise was more than 10 times higher than single scattering. Even if our algorithm raised single scattering intensity at the confocal points by 400 times, the apparent increase of image intensity and the resulting contrast enhancement was only about 30 times. We think that this is another reason why the reviewer considered the enhancement rather moderate. But this is another innate difficulty arising in the reflectance imaging.

We added the following paragraph to the revised manuscript to explain the detrimental effect time-gated multiple scattering noise to the improvement of image quality in the reflectance imaging.

“From the obtained pupil aberration maps, it is estimated that the PSF width of the aberrated single-scattered waves is about 6-8 μm in full width at half maximum, which causes a reduction in the peak intensity of single-scattering signal in the confocal spots by a factor of ~ 400 . By comparing the PSFs before and after aberration correction, we estimate that the ratio of single-scattering signal to time-gated multiple-scattering background noise at confocal points was initially about 0.08, much smaller than 1, before the aberration correction. This explains why the conventional OCM failed to achieve high-resolution imaging of mouse brain. The CLASS algorithm selectively refocused the aberrated single-scattering signals back to the confocal points to raise the single scattering intensity by a factor of ~ 400 . This made the single scattering intensity larger than time-gated multiple scattering noise by about 30 times after the aberration correction and enabled us to identify individual myelinated axons with diffraction-limited resolution (see Supplementary Section VIII for detailed PSF analysis).”

The reviewer raised an interesting point that the use of higher excitation power may lead to the comparable increase of image quality to that of the aberration correction. We realized that the addressing of this point substantiates how critical it is to deal with the sample-induced aberrations in the label-free reflectance imaging. In the following, we made it clear that the spatial resolving power and signal to noise ratio (SNR) of the OCM imaging cannot be improved simply by raising the excitation laser power. The increase of excitation power can be helpful only when signal is so weak as to be comparable to or smaller than the shot noise/detector noise (camera's dark noise and read-out noise). In the reflectance imaging, however, multiple scattering noise and non-confocal signals, responsible for distortion of the illumination and detection PSFs, are well above the limits of photon shot noise and detector noise. Therefore, the increase of excitation power results in the same speckle noise as that of the weaker excitation.

To directly prove this point, we obtained OCM reflectance images through the 90- μm -thick intact skull from a 4-week-old transgenic mouse for various illumination powers (Figure R2a-c). The detector (camera) was saturated at a power of 15 mW at the sample. In addition, we measured 25 images independently at 15 mW illumination and averaged them all (Figure R2d), which is equivalent to the excitation of 375 mW. As shown in Figure R2a-d, OCM reflectance images did not change at all, including even the micron-structured speckle-like patterns, regardless of the illumination power because the detected signal was already well above the limits of shot noise and detector noise. There is no improvement in the image quality, and any distinct features such as myelin fibers were not recovered at all. Myelinated axons can only be visible when the broadened non-confocal signals are refocused back to the confocal position by means of aberration correction.

Figure R2. OCM and TPF images depending on the illumination power. **a-c**, OCM images by varying the illumination power at the same position of the mouse brain tissue under the intact skull as shown in Fig. 4. The beam intensities measured at the sample plane were 2.2 mW, 6.6 mW and 15 mW, respectively. The color bar, intensity normalized by the maximum intensity in **a**. **d**, Averaged image of 25 OCM images measured at the illumination power of 15 mW. **e-g**, TPF image obtained by increasing the illumination power. Color bar, intensity normalized by the maximum intensity in **e**. **h**, Averaged image of 25 TPF images taken at the excitation power of 15 mW.

The reviewer's reasoning is partly true for multi-photon imaging. When the peak intensity of the excitation PSF is attenuated due to multiple scattering and aberration, TPF signal can be too weak to ignore shot noise/detector noise. In such case, the increase of the excitation power can enhance SNR. To complete this discussion, we took TPF images for the same area as those in Figure R2a-d. Here, TPF signal reports neuronal dendrites as the transgenic mouse used in this experiment expresses EGFP at the neuronal membranes. Multiple TPF images were taken while increasing the excitation power (Figure R2e-h). In contrast to the OCM reflectance images, the SNR of TPF images was increased because stronger excitation compensates the attenuation of the excitation PSF by the skull and raises fluorescence signals above shot noise/detector noise limits (Note that strong elastic backscattering of excitation, serving as multiple scattering noise in the reflectance imaging, is filtered out in the fluorescence imaging by using a dichroic mirror and color filters). However, the resolving power of TPF images was far from the diffraction limit because the PSF broadening due to aberrations remained to be dealt with. It is also noteworthy that TPF imaging shows much better image contrast than reflectance OCM imaging when both are subject to the skull-induced aberrations.

For the case of through intact skull imaging, the imaging results were not consistent. If we compare Fig. 3L and 3M, some of the features that are available in L were not visible in M while some features of M were not visible in L. Moreover, there is no ground truth for this measurement. I assume one can remove the skull and image the same area and provide a ground truth image of the axon.

The thickness of the skull was $120\ \mu\text{m}$ in Figs. 3k-m, where the isoplanatic patch size was extremely small. According to our analysis, the isoplanatic patch was as small as $7.5\times 7.5\ \mu\text{m}^2$. As explained in the main text, we applied CLASS algorithm to each subregion to cope with spatially varying aberrations. If the subregion is larger than the isoplanatic patch, then the aberration varies significantly within the subregion. CLASS algorithm tends to find the average of spatially varying aberrations, which resulted in the loss of high-order modes of aberration. Too much of averaging can lead to the complete loss of resolving power in the worst case. For example, the size of subregion was $30\times 30\ \mu\text{m}^2$ in Fig. 3k, much larger than the isoplanatic patch size. No structures were resolved in this case due to the significant averaging of spatially varying aberrations. As we gradually reduced the size of subregion to $15\times 15\ \mu\text{m}^2$ (Fig. 3l) and $7.5\times 7.5\ \mu\text{m}^2$ (Fig. 3m), we could observe the appearance of filamented structures, the signature of myelinated axons.

We have a physical criterion for the optimal size of subregion. CLASS algorithm is intended to constructively accumulate non-confocal signal back to confocal points. If it works well, then the total intensity in the reconstructed image is to be increased. We plotted the total intensity of the CLASS image as a function of the size of subregion in Figure R3d. From this plot, we learned that the subregion size of $7.5\times 7.5\ \mu\text{m}^2$ is the best condition. Further reduction in the subregion size makes the reconstruction more susceptible to the multiple scattering noise.

Regarding the issue of the ground truth, we acknowledge that it is extremely difficult to compare our results with the ground truth because it is almost impossible to acquire images without the skull. The imaging condition, i.e. reflectance imaging under $120\text{-}\mu\text{m}$ -thick skull in vivo, is too stringent to obtain similar results by other imaging modalities. This is why we presented thinned skull imaging (Figs. 3b-d) and relatively thin intact skull imaging (Figs. 3f-h), where conventional OCM imaging can serve as ground truth to some extent. We think that it is rational to demonstrate new methodology in the regime where the other methods are marginally working and then to trust new results at the extreme regime where there is no other way to reach.

We considered the reviewer's suggestion of taking images at the same area after removing the skull, but it was technically too challenging. Instead, we performed adaptive optics TPF imaging with the aberration maps identified by LS-RMM and demonstrated near-diffraction-limited TPF imaging of dendritic spines, a type of ground truth, through an intact skull (Fig. 4 in the revised manuscript and our response to the following comment).

Figure R3. Finding the isoplanatic patch by varying the size of the subregion. **a**, CLASS intensity image of in vivo mouse brain with the intact skull for a $30\times 30\ \mu\text{m}^2$ area. **b-c**, Intensity images with CLASS applied independently to subregions divided into 2×2 and 4×4 , respectively. **d**, Total intensity in the same view field as **a** depending on the side length of the subregion. The point indicated by the

red arrow, at which the total intensity becomes the maximum, corresponds to **c**.

To clarify the meaning of the analysis in Figs. 3L and 3M, we added the following sentence to the main text and added the new analysis similar to Fig. R3 to the Supplementary Section VII.

“The isoplanatic patch size can systematically be found by finding the size of subregion at which the intensity of the aberration-corrected image is maximum (see Supplementary Section VII).”

In addition, there is not SHG imaging for the through intact skull imaging measurement. If the correction is effective, it should at least show large signal increases (second order signal).

We already proved the effectiveness of the hardware aberration correction of SHG imaging in the Supplementary Information in the original manuscript. We demonstrated the recovery of the collagen fiber structures located under an intact skull. In this revision, we applied LS-RMM for the adaptive optics TPF imaging and demonstrated near-diffraction-limited TPF imaging of neuron's dendrites through an intact skull (Figure R4). We believe that the addition of this new results strongly supports the validity of LS-RMM. Brief summary of the results is given below (see details in Fig. 4 and main text of the revised manuscript).

We prepared a fixed whole mouse brain including an intact skull from a 4-week-old transgenic mouse that expresses enhanced green fluorescent proteins (EGFP) at the neuronal membranes. We first performed CLASS imaging to obtain the aberrations induced by the mouse skull. Objective focus was set to 125 μm below the upper surface of the skull, whose thickness was about 85 μm . We divided the entire field of view into 14×14 subregions and applied the CLASS algorithm for each subregion to recover fine myelination structures therein (Figure R4b) and to obtain the aberration map at the corresponding subregion (Figure R4c). The number of correction modes in each aberration map was 9,880.

Similar to the demonstration in Figs. 2j-m, we physically corrected the skull-induced aberrations by displaying the phase conjugations of the aberration maps on the SLM. Since the aberration varied depending on the position, we made a physical correction for each subregion at a time. As a point of reference, a conventional TPF image was taken without aberration correction. Figure R4d shows a maximum intensity projection (MIP) of a volume image obtained in a depth range of 119-135 μm over the same field of view as that in Figure R4a. The contours of neuronal dendrites are blurry, and their microstructures are invisible due to the significant broadening and attenuation of the excitation PSF. For the subregion marked by the yellow dotted box in Figure R4d, we applied the hardware correction using the aberration map obtained from the same subregion in Figure R4b and conducted the TPF imaging. The TPF image with hardware aberration correction (Figure R4e) shows 19 times increase in fluorescence intensity and enables the recovery of the fine dendritic spines that were invisible before the correction. Notably, only the structures in and around the subregion were corrected properly, supporting that the isoplanatic patch size is about the same as the subregion indicated by the yellow box, which is $10 \times 10 \mu\text{m}^2$. For the full mapping of the neuronal dendrites, we conducted TPF imaging for each subregion indicated by the dotted gray box in Figure R4b by displaying the phase conjugation of the associated aberration map in Figure R4c. Multi-depth TPF images were obtained by scanning the imaging depth over the same depth range in Figure R4d, where the imaging depth was scanned by adding a defocus phase map to the SLM. Figure R4f and 4g show MIP images for a depth range of $113 \pm 1.5 \mu\text{m}$ before and after aberration correction, respectively. Figure R4h and Figure **R4i** are the same as Figure R4f and Figure **R4g**,

respectively, but for a depth range of $122\pm 1.5\ \mu\text{m}$. The corrected view field was expanded as expected, and dendrites and spines at two different depths were clearly resolved. The measured width of dendritic spine was as small as 500 nm, close to the diffraction limit.

We added these new results to Fig. 4 of the revised manuscript. In addition, the following paragraphs were added to clarify the benefits of LS-RMM in the context of adaptive optics multi-photon imaging.

“TPF imaging that we demonstrated here can be considered a type of wavefront sensing AO. LS-RMM serves as a tool to measure wavefront distortion by scattering tissues, similar to Shack-Hartmann wavefront sensors and other interferometric microscopy techniques. The uniqueness of our approach is that LS-RMM can identify extremely severe skull-induced aberrations without using either external guide stars or nonlinear fluorescence excitation. This benefit comes from the recording of the time-resolved reflection matrix and the use of CLASS algorithm extracting the one-way aberration. In LS-RMM, the identification of wavefront aberrations is based on the intrinsic reflectance contrast. Myelinated axons are good source of intrinsic contrast as they are spread throughout the mouse brain. In fact, it is not always necessary for the myelinated axons to be around the region of interest for multi-photon imaging. Even for the area where there are no axons such as the subregion indicated by the white arrowhead in Fig. 4b, we were able to identify the aberration map and obtain near-diffraction-limited TPF images. This is because noise-like speckle patterns backscattered from irregular tissue structures at the focal plane are also single-scattered waves contributing to the CLASS algorithm.

Adaptive optics by LS-RMM is beneficial over existing AO modalities that rely upon feedback optimization of multi-photon fluorescence image in that photobleaching effect is negligible during the measurements of the aberrations as it does not require high excitation power to record the reflection matrix. We used an excitation power of 21 mW at the sample and a pixel exposure time of 100 μs for TPF imaging, whereas an excitation power of 3 mW was used for recording the reflection matrix. Another important benefit is that recording of one reflection matrix over the area of $150\times 150\ \mu\text{m}^2$ enabled us to obtain 15×15 number of aberration maps for as many different isoplanatic patches. In the iterative feedback AO, multiple iterations of optimization need to be conducted for each isoplanatic patch.”

Figure R4. Two-photon fluorescence imaging through an intact mouse skull. **a**, Conventional OCM image under an intact mouse skull before aberration correction. The thickness of the skull was about 85 μm , and focal plane was set to a depth $z_0 = 125 \mu\text{m}$ from the upper surface of the skull. **b**, LS-RMM image stitched after applying aberration correction to each of 15×15 subregions. **c**, Aberration maps of the subregions indicated by the gray dotted line in **b**. The size of subregion is $10 \times 10 \mu\text{m}^2$, and each phase map contains 9,880 angular modes. Color bar, phase in radians. **d**, TPF image at the same position as **a** before hardware aberration correction. The image was obtained by the maximum intensity projection for a depth range of 119-135 μm with 0.5- μm spacing. **e**, TPF image after physical aberration correction for the subregion indicated by the yellow box in **b**. Yellow boxes in **d** and **e** correspond to the same yellow box area in **b**. **f** and **g**, MIP of TPF images at the depth $z_1 = 113 \pm 1.5 \mu\text{m}$ before and after aberration correction, respectively, for the area indicated as white dashed box in **d**. **h** and **i**, Same as **f** and **g**, respectively, for the depth $z_2 = 122 \pm 1.5 \mu\text{m}$. Color bar, intensity normalized by the maximum intensity in **i**. Scale bars indicate 30 μm in **a**, **b** and **d**, and 10 μm in **e-i**.

Overall, the data seems to suggest that this method works better for aberration (not very scattering) sample but seems to have trouble working with scattering sample. The quality of the through-skull imaging is not great. It seems that regular OCM or SHG with a bit higher power can achieve the same image quality for the case of thinned skull imaging. The through skull imaging has no ground truth and no SHG image comparison. So it is hard to tell if the method indeed yields correct images.

We addressed all these points in our response to the previous comments.

I'd suggest that

- 1) provide ground truth image for comparison. Otherwise, we can not tell if the images were correct for the case of through skull imaging;*
- 2) provide SHG image comparison for through intact skull data. A large signal increase in SHG can demonstrate the effectiveness of the correction.*

Once again, the newly added results demonstrating the recovery of the diffraction-limited spatial resolution in TPF imaging by using the aberration maps acquired by LS-RMM strongly support the validity of the proposed method and clearly resolve the ground truth issue.

- 3) more and better in vivo test results. Is the system really reliable and working well in most cases (so that people can use it consistently)?*

Mouse skull presents extreme form of aberrations. The required number of correction modes amounts to more than 1000, well beyond the capability of the conventional adaptive optics. Furthermore, the aberrations are spatially varying to the degree that even 10 μm shift of the view field results in the completely different aberrations. All these complications have made it nearly impossible to perform label-free reflectance imaging under an intact mouse skull so far. We rigorously proved that our method can deal with all these challenges and demonstrated the diffraction-limited imaging for both reflectance imaging and multi-photon imaging through an intact skull. Based on our experience, our method is highly reproducible up to the skull thickness of 100 μm . Further increase in the skull thickness makes the problem extremely challenging. The aberration map in Fig. 3m is close to speckle, where the distinction between aberration and multiple scattering becomes subtle. On this condition, experiments become highly susceptible to various factors such as the motion of the specimens and the degree of multiple scattering noise. We think that the use of higher speed camera will make our system more reliable for in vivo imaging, which is one of our future directions.

We added the following sentences to discuss the reliability in the in vivo imaging.

“Since the aberration correction takes place computationally in the coherent reflectance imaging, the recording of the reflection matrix itself is the end of imaging session. This is particularly beneficial for in vivo imaging because we have ample time to correct severe position-dependent aberrations after the data acquisition. Still, the data acquisition speed is critical for in vivo imaging because even small perturbation for the reflection matrix can undermine the image reconstruction, especially in the case of severe aberrations.”

- 4) discussion for the performance difference between the test target imaging (through rough plastic surface) and the through skull imaging (real biological scattering tissue).*

As we explained above, the contrast of the reflectance images of myelinated axons are intrinsically low in comparison with the test target. Therefore, the performance of our method for deep-tissue imaging appears to be lower than that of the test target images. However, this doesn't mean that our reconstruction cannot deal with real biological scattering tissues. In fact, our algorithm worked well based on the fact that the reconstructed image through an intact skull shows similar contrast with those taken without aberrations and multiple scattering.

When all the other conditions are the same, then the use of targets with high contrast will allow us to go deeper inside. The CLASS algorithm is to find the correlation among single scattering in the background of multiple scattering noise. Therefore, it is likely that single scattering from stronger contrast targets will provide better correlation for the same scattering noise. In biological imaging, however, we have no choice but to use the intrinsic targets. Fortunately, we could cope with the multiple scattering generated by the mouse skull and 100 μm -thick tissues by using the intrinsic reflectance contrast. Further increase in depth generates too strong multiple scattering noise for the CLASS algorithm to keep up with. We expect that the use of longer excitation wavelength will lead us to imaging deeper due to reduced multiple scattering noise.

We added the following sentences to the discussion section of the revised manuscript.

“The working depth of current implementation is set by the weak reflectance contrast of myelinated axons relative to the multiple scattering noise. The use of exogenous contrast agents may increase the imaging depth, but at the expense of perturbing normal physiology. The use of longer excitation wavelength can potentially be helpful as the multiple scattering noise is to be reduced.”

Author's Response to Reviewer #2

In this work, the authors investigate a reflection matrix approach of label-free optical imaging that allows to overcome high-order spatially-varying sample-induced aberrations. In particular, the authors demonstrate diffraction-limited imaging of myelinated axons through ex-vivo mouse skulls. Compared to conventional adaptive optics, a reflection matrix approach does not require any guide star and allows to tackle space variant aberrations. Once the reflection matrix is recorded, the CLASS algorithm, developed in previous works by the same authors and improved in that paper, allows to correct in post-processing high-order aberrations both at input and output. The main novelty in the current paper is the measurement of the reflection matrix in a focused basis both at input and output, while, in previous papers, the authors were measuring the reflection matrix in a plane wave basis. Albeit more challenging due to shot-to-shot phase fluctuations, the focused basis ensures, according to the authors, "the optimal use of the detector's dynamic range, resulting in enhanced sensitivity to multiple scattering noise". This is an aspect with which I may disagree and that I will discuss below.

In overall, this paper is of excellent quality, well written, concise and clear. This work is useful both for researchers working in the field of wave-front shaping and those more generally motivated by the difficult and long-standing challenge of imaging in/through complex media. I think it perfectly fits with the scope of Nature Communications. For all these reasons, I would suggest to accept this paper. However, one main objection that can be addressed to the authors is the question of novelty of this work with regards to their own previous papers also published in Nature Communications [27,28]. I would suggest the authors to be clearer about the merit of this current work compared to previous ones. Otherwise, one may think this work is only incremental.

We deeply appreciate the reviewer's thinking highly of the quality and the importance of our work. We addressed the reviewer's valuable comments and suggestions in this response letter and the revised manuscript.

For the past few years, we have been reporting noteworthy methodologies for deep optical imaging within complex scattering medium. In our earlier work in 2015 (Ref. 30), we measured the time-gated reflection matrix for the first time and developed an algorithm termed collective accumulation of single scattering (CASS) that efficiently extracts the object function under strong multiple scattering background. In the following study in 2017 (Ref. 27), we proposed new algorithm, the closed-loop accumulation of single scattering (CLASS), that addresses multiple scattering and sample-induced aberrations all together. In the study reported in 2019 (Ref.29), we increased the data acquisition speed of the time-gated reflection matrix in the planar wave basis and realized the imaging of central nervous systems of a living zebrafish.

In the present study, we made a significant improvement in both instrumentation and algorithm to cope with the extremely complex aberrations induced by the intact mouse skull. We demonstrated label-free reflectance imaging through intact skull for the first time to our knowledge and demonstrated near-diffraction-limited two-photon fluorescence imaging through an intact skull by the hardware aberration correction.

In terms of instrumentation, we devised the laser scanning reflection-matrix microscope (LS-RMM) for measuring the time-gated reflection matrix by the focused illumination and image-plane detection. As we discussed in the following, this configuration allows us to optimally use the detector dynamic range for detecting severely aberrated single scattering signal embedded within multiple scattering noise. Furthermore, this configuration is completely compatible with

multi-photon microscopy. We could apply the aberration map identified by LS-RMM to two-photon fluorescence imaging and SHG imaging through intact skull by simply inserting an SLM in the illumination beam path. Regarding the software, we made significant improvement of the CLASS algorithm in such a way to deal with extremely small isoplanatic patch size without losing the spatial frequency resolution of the aberration correction (Supplementary section VI).

These points were made clear in the introduction of original manuscript, but we streamlined the introduction for further clarification. We also add the following sentence to emphasize new experimental results demonstrating hardware AO for two-photon fluorescence imaging.

“We realized HAO by displaying the conjugation of the aberration maps identified from the reflection matrix by simply inserting the SLM in the illumination beam path and demonstrated AO multi-photon imaging through an intact mouse skull. In particular, we performed two-photon fluorescence imaging of the neuronal dendrites and visualized their spines with the spatial resolution of 500 nm, close to the diffraction-limited two-photon imaging resolution of 380 nm, over the field of view much larger than the isoplanatic patch.”

1. With respect to this novelty issue, I am questioning the interest of recording the reflection matrix in a focused basis both at emission and reception rather than in a plane wave basis. I am not sure to understand why this basis allows the optimal use of the detector’s dynamic range. In a focused basis, most of the reflected wave-field is localized in the vicinity of the input focusing point while it spreads over the whole detector surface when the latter one is placed in the pupil plane. If we have a good photon budget, this last configuration seems to be better in terms of sensitivity since more photons can be detected before reaching the saturation of the camera. Could the authors comment on that?

We agree with the reviewer’s opinion in the case of weak scattering/aberration regime. The backscattered waves, mostly composed of single-scattered waves, are localized around the focusing point in the image plane. As the reviewer commented, only a small number of pixels in the vicinity of the focusing point contribute to the detection dynamic range in the focused basis. In contrast, the reflected wave is more evenly distributed over the entire pupil plane. One can make use of the dynamic range by larger number of pixels in the pupil-plane detection. However, there is no detector dynamic range issue in the weak scattering/aberration regime. Recording at a few camera pixels is good enough to acquire high fidelity images as long as most of their dynamic range is used for recording the single-scattered wave. In fact, confocal detection is an extreme case, where only a single pixel detection is used for imaging.

The image-plane detection becomes beneficial over the pupil-plane detection as multiple scattering noise is increased. In the extreme multiple scattering and aberrations, the camera pixels are saturated predominantly by strong multiple-scattering light regardless of the location of the detection plane. In the pupil-plane detection, all the multiple scattering collected by the objective lens uniformly fill the pupil aperture. In the image-plane detection, the collected multiple-scattering light spreads over the full field of view of the objective lens (FOV_{obj}), which is much larger than our FOD at the camera. FOV_{obj} of our microscope (limited by 1-inch-diameter eyepiece lens) is about 250 μm in diameter, and the FOD was $50 \times 50 \mu\text{m}^2$ for the through-skull imaging. Therefore, only a factor of 25 of total collected multiple scattering is detected in the image-plane detection (see Supplementary Section IV for details). When we convert the image-plane basis to the pupil-plane basis for the CLASS algorithm, the single- to multiple-scattering intensity ratio is reduced by the same factor in comparison with the pupil-plane detection.

Another important benefit of the image-plane detection is that the weak single scattering signal is localized and gathered together in space. The single scattering intensity at each pixel is higher than that in the pupil-plane detection. Therefore, the ratio between single- and multiple-scattered waves at the camera pixel is higher in the image-plane detection. This means that the camera can detect single scattering at a deeper imaging depth in the image-plane detection than in the pupil-plane detection.

Finally, the matrix acquisition time can be shortened in the image-plane detection. As discussed in the manuscript, the optimum size of FOD is determined to be wide enough to capture the entire PSF broadened by the sample-induced aberrations. We can adaptively adjust the size of FOD at the camera with the increase of imaging depth in such a way to optimize matrix acquisition speed. This is especially critical for in vivo imaging where the motion of the living specimen can undermine the image reconstruction process.

We added the additional benefit of image-plane detection in the matrix acquisition time to the introduction section.

“Furthermore, the matrix acquisition time can be shortened in the new configuration. The view field for recording non-confocal signals needs to be wide enough to capture the entire PSF broadened by the sample-induced aberrations. We can adaptively adjust this view field at the camera with the increase of imaging depth in such a way to optimize matrix acquisition speed. This is especially critical for in vivo imaging where the motion of the living specimen can undermine the image reconstruction process.”

2. A second advantage of the focused basis invoked by the authors is the fact that “signals” can be more easily discriminated from the multiple scattering noise than in a plane wave basis. Indeed, the single scattering contribution is more likely to be focused in the vicinity of the input focusing point, while the multiple scattering noise spreads over the whole image plane. I agree but one could have recorded the reflection matrix in the pupil plane, project the data in the focused basis via a numerical Fourier transform (as the authors do) and apply an adaptive confocal filter on the data in the focused basis in order to restrict their field of illumination and/or detection as needed (as done in Ref.32). The reflection matrix in a plane wave or focused basis carries the same information. One can easily project the wave-field from one basis to another, depending on what we want to do (aberration correction, multiple scattering filter, etc.). Hence there does not seem to have much interest of recording the reflection matrix in a focused basis rather than in a plane wave basis, like the authors did in their previous works.

We completely agree with the reviewer in that a reflection matrix measured in one basis can be transformed into another basis by the post processing. We also agree that the single-scattering filter (SS filter) used in Ref. 32 can be applied after the proper basis transform to the space domain. However, this is true only when the detector has enough dynamic range to properly record single scattering at the camera. In fact, the FOD in LSRMM is the hardware version of SS filter used in Ref. 32, a loose confocal gating that efficiently discriminates multiple-scattering noise from single-scattering (or snake-like) signal. In contrast to the SS filter that discriminates the “already detected” multiple-scattering noise, the hardware filter provided by LS-RMM is advantageous in making use of detector dynamic range because the single- to multiple-scattering ratio is higher in the recording stage as discussed in the comment #1.

The efficient use of detector dynamic range is crucial not only for increasing the SNR, but also for reducing data acquisition time, which is essential for in vivo bioimaging applications. High-

speed camera tends to have low dynamic range due to the demand for the data transfer speed. The image-plane detection can allow us to use high-speed camera while maintaining enough SNR.

We added this discussion on the advantages of the image-plane detection to Supplementary Section IV.

3. The main improvement on the CLASS algorithm proposed in that paper is the possibility of considering a field-of-detection (FOD) much larger than the field-of-illumination (FOI). Indeed, for a local correction of position-dependent aberrations, the FOI should be restricted to an isoplanatic patch while the FOD should capture the whole imaging PSF. The latter parameter governs the number of angular modes over which aberrations can be corrected. While the authors discuss in depth the choice of the FOD as a compromise to be found on the signal-to-multiple scattering ratio, the choice of the FOI is not discussed at all. How do they a priori determine the size of an isoplanatic patch? How do they choose the sub-division of the field-of-view?

This is an interesting point that warrants detailed discussion. Since we measured the reflection matrix over the FOI larger than the isoplanatic patch, we can computationally choose the size of the subregion within the FOI that best matches the sample's isoplanatic patch. When the subregion is larger than the isoplanatic patch, aberration varies within the subregion. Then the correlation between the pupil functions for different locations within the subregion becomes lower. As a consequence, high-order aberrations in the pupil tend to be averaged out, and relatively low-order aberrations can only be found. This was made clear in the aberration-corrected CLASS images for different choices of the subregion size in Fig. 3k-m. For the subregion size of $30 \times 30 \mu\text{m}^2$ in Fig. 3k, aberration map was too smooth to recover microscopic structures of myelinated axons. As we gradually reduced the size of subregion to $15 \times 15 \mu\text{m}^2$ (Fig. 3l) and $7.5 \times 7.5 \mu\text{m}^2$ (Fig. 3m), fine aberration maps were found. When the size of subregion is too small, the number of input modes becomes too small to overcome the disturbance by the multiple scattering noise.

We can systematically find the optimal subregion size by monitoring the total intensity of the CLASS image. The CLASS algorithm is intended to constructively accumulate non-confocal signal back to confocal points. If it works well, then the total intensity in the reconstructed image is to be increased. To make this point clear, we chose $30 \times 30 \mu\text{m}^2$ area in the intact skull imaging in Fig. 3f, which is shown as dotted box in Figure R5a. And we applied CLASS algorithm while reducing the size of subregion (Figure R5b-f) and plotted the total intensity of the CLASS image as a function of the size of subregion in Figure R5d. We found that the total intensity is maximum when the subregion size is $10 \times 10 \mu\text{m}^2$, where the filamented structures of myelinated axons are clearest.

We added this analysis for systematic search for the isoplanatic patch size to Supplementary Section VII.

Figure R5. Systematic search for the isoplanatic patch size. **a**, Conventional OCM imaging through the intact skull of an 8-week-old mouse shown in Fig. 3f. **b-f**, CLASS images obtained by varying the size of the subregion (L_{sub}) for a $40 \times 40 \mu m^2$ indicated by the white box of **a**. L_{sub} 's corresponding to **b-f** are 40 μm , 20 μm , 10 μm and 5 μm , 2.5 μm , respectively. **g**, Graph showing the relationship between L_{sub} and total intensity of **b-f**. The point with the largest total intensity indicated by the red arrow corresponds to **d**, $L_{sub} = 10 \mu m$.

Besides these main points, I have few minor comments:

4. The authors mention that they can remove most of the multiple scattering background by carefully choosing the FOD but they do not provide any quantification of the multiple scattering level in their experiments. Related to that question, do they have an estimation of the scattering or transport mean free paths in the mouse skull and brain such that they can express the imaging depth in terms of scattering/transport mean free path?

The quantification of multiple scattering is rather complicated issue especially because the distinction between the aberrated single scattering and time-gated multiple scattering is often subtle. We compared PSFs before and after aberration correction and extracted single- and multiple-scattered waves to the best estimation possible. Figure R6 shows the intensity PSFs before and after aberration correction obtained from the through-skull imaging of 8-week-old mouse brain in Fig. 3e-h. The width of PSF broadened due to aberration and multiple scattering was narrowed down by 15 times to form a near-diffraction-limited PSF after the aberration correction. The peak intensity at the center of the PSF was increased by about 30 times.

The PSF before aberration correction consists of the aberrated single-scattered wave and time-gated multiple-scattered wave, i.e. $E_{lab}(\mathbf{r}_o; \mathbf{r}_i) = E_S(\mathbf{r}_o; \mathbf{r}_i, \tau = \tau_0) + E_M(\mathbf{r}_o; \mathbf{r}_i, \tau = \tau_0)$. After the aberration correction, the aberrated single-scattered wave is focused back to $\mathbf{r}_o = \mathbf{r}_i$, the original illumination spot. If we denote the Strehl ratio enhancement of the single-scattered wave as α_s , then aberration-corrected field can be written as $E_{lab}^c(\mathbf{r}_o = \mathbf{r}_i; \mathbf{r}_i) = \sqrt{\alpha_s} E_S(\mathbf{r}_i; \mathbf{r}_i, \tau = \tau_0) + E_M(\mathbf{r}_i; \mathbf{r}_i, \tau = \tau_0)$. In our experiment, we can estimate α_s from the aberration maps identified by our CLASS algorithm. In the case of skull imaging (Fig. 3h), α_s is estimated to be about 400. Therefore, the increase of peak intensity after the aberration correction is written as

$$\alpha_{S+M} = \frac{\alpha_s |E_S|^2 + |E_M|^2}{|E_S|^2 + |E_M|^2}.$$

In our experiment in Fig. R6c, α_{S+M} was measured to be about 30. Therefore, $\frac{|E_S|^2}{|E_M|^2} \approx \frac{\alpha_{S+M}}{\alpha_s - \alpha_{S+M}}$ was about 0.08, which means that single-scattered wave was more than 10 times weaker than the time-gated multiple-scattered waves. Considering that multiple scattering with different flight times is about 10 times stronger than the time-gated multiple scattering (see Supplementary Section IV), total multiple scattering was initially more than 100 times stronger than single scattering at the confocal points.

The enhancement of the Strehl ratio is often used as a measure of the performance of an AO system because it indicates the increase of reconstructed image intensity after the aberration correction. In the weak multiple scattering, the Strehl ratio enhancement is determined by α_s . On the contrary, it is measured to be $\alpha_{S+M} (\ll \alpha_s)$ as multiple scattering is significantly larger than the single scattering. Therefore, even if the aberrated single-scattered wave is well focused back to the confocal point, the effective Strehl ratio enhancement, α_{S+M} , appears to be much lower due to the presence of multiple scattering noise.

Figure R6. PSFs before and after aberration correction for the through-skull imaging. **a**, Intensity PSF before aberration correction. **b**, Intensity PSF after aberration correction by CLASS algorithm. The color bars are normalized by the maximum value in **b**. **c**, Line profiles of PSFs before (blue) and after (red) aberration correction. The PSF intensity before aberration correction is enlarged by 20 times for visibility.

The measurement of the scattering mean free path l_s of the mouse skull is tricky as we cannot easily control its thickness. Moreover, the skull consists of several layers of microstructures having different optical properties. Still, we can make a rough estimation. We placed a flat mirror under an excised skull and measured the intensity of the ballistic photons in the reflection side. We measured the attenuation of the intensity of the ballistic photons with respect to the intensity measured by the mirror itself, i.e. I_B/I_0 , and the thickness, L , of the mouse skull from the SHG imaging. we can determine the average scattering mean free path by the relation $I_B = I_0 \exp(-L/l_s)$. According to our measurements for the skulls with various thicknesses, l_s was estimated to be about 30-40 μm , similar to the value measured in Ref.10. The scattering mean free path of the cortical brain tissues was measured to be about 100 μm from the same measurements. From these measurements, the imaging depth of our through-skull imaging is about $4l_s$. This may appear to be moderate. However, we need to consider that the degradation of Strehl ratio by the severe aberrations of the mouse skull is more than a factor of 1/100 and the reflectance of the myelinated axons is less than 0.01.

Including these factors, the effective imaging depth corresponds to more than $8 l_s$ in comparison with the ideal case when a scattering phantom is placed on the resolution target.

We added the following paragraph to the revised manuscript to explain the PSFs before and after the aberration correction.

“From the obtained pupil aberration maps, it is estimated that the PSF width of the aberrated single-scattered waves is about 6-8 μm in full width at half maximum, which causes a reduction in the peak intensity of single-scattering signal in the confocal spots by a factor of ~ 400 . By comparing the PSFs before and after aberration correction, we estimate that the ratio of single-scattering signal to time-gated multiple-scattering background noise at confocal points was initially about 0.08, much smaller than 1, before the aberration correction. This explains why the conventional OCM failed to achieve high-resolution imaging of mouse brain. The CLASS algorithm selectively refocused the aberrated single-scattering signals back to the confocal points to raise the single scattering intensity by a factor of ~ 400 . This made the single scattering intensity larger than time-gated multiple scattering noise by about 30 times after the aberration correction and enabled us to identify individual myelinated axons with diffraction-limited resolution (see Supplementary Section VIII for detailed PSF analysis).”

5. Line 196: The authors say “The sample-induced aberrations are dominated by angle-dependent phase retardation”. Why? Is it because the skull is far away from the focal plane and can thus be considered as a far-field phase screen?

This sentence can be misleading, and we revised the text as “In order to apply the CLASS algorithm, we converted...”

Adaptive optics microscopy usually places a wavefront shaping device at the pupil plane. This means that the phase stroke in each pixel in the device controls the relative phase of the angular planar wave incident to and/or reflected from the sample. That is why the sample-induced aberrations are mostly considered the angle-dependent phase retardations. In CLASS algorithm, we follow the same convention and identify the angle-dependent phase retardations in the wavevector space. As we will discuss in comment #6 in the following, the correction of the angle-dependent phase retardations is not perfect when the aberrating layers are thick and extended to the vicinity of the focal plane. This is why the isoplanatic patch size is finite in the through-skull imaging and multiple position-dependent aberration maps are required for proper correction. As we demonstrated in our main text, our improved CLASS algorithm can deal with extremely small isoplanatic patch sizes without losing the angular resolution of the correction.

6. In relation to the previous point, is the CLASS algorithm limited angle-dependent phase retardation? In other words, can it also address space-dependent phase retardation effects induced by sample inhomogeneities close to the focal plane?

This is exactly what we demonstrated in the through-skull imaging (Fig. 3). The improved CLASS algorithm demonstrated in the present study can deal with the space-dependent phase retardations induced by the sample inhomogeneities close to the focal plane. In the *in vivo* imaging, the focal plane was placed 40 μm from the bottom surface of a 120- μm -thick skull. Therefore, the skull was in the proximity of the focal plane. The skull cannot be considered a phase plate in the far-field, and its isoplanatic patch size is extremely small. Our algorithm could keep up with the patch size as small as $7.5 \times 7.5 \mu\text{m}^2$.

7. Line 198: Why does the projection of the reflection matrix in the plane wave basis writes: $\tilde{\mathbf{R}} = \mathbf{F}^{-1} \mathbf{R} \mathbf{F}$? According to me, the Fourier transform should have the same convention at the input and output: $\tilde{\mathbf{R}} = \mathbf{F}^T \mathbf{R} \mathbf{F}$ (where the symbol T stands for transpose). Otherwise, the reflection matrix is no longer symmetrical while it should be by virtue of spatial reciprocity.

We appreciate the reviewer's point out this issue. In fact, we made a mistake in our basis transformation expression. $\tilde{\mathbf{R}} = \mathbf{F} \mathbf{R} \mathbf{F}^{-1}$ is a correct expression, not $\tilde{\mathbf{R}} = \mathbf{F}^{-1} \mathbf{R} \mathbf{F}$. We used the correct one in our experimental data analysis. Regarding the difference between $\tilde{\mathbf{R}} = \mathbf{F} \mathbf{R} \mathbf{F}^{-1}$ and $\tilde{\mathbf{R}} = \mathbf{F} \mathbf{R} \mathbf{F}^T$, this difference is only a matter of the choice of the coordinate system. Let us consider a planar wave with the transverse wavevector $(k_x^i \neq 0, k_y^i = 0)$ is incident to a flat mirror as shown in Figure R7a. In the reviewer's preference, $\tilde{\mathbf{R}} = \mathbf{F} \mathbf{R} \mathbf{F}^T$, the sign of the transverse wavevector of the reflected wave is reversed, i.e. $k_x^o = -k_x^i$ because the x -axis of the coordinate system describing the reflected wave is reversed to preserve the handedness of the Cartesian coordinate system (Figure R7b). This is similar to the reflection of the circular polarization. Right-handed circular polarization becomes left-handed circular polarization upon the reflection by the mirror due to the coordinate system. In our choice of transformation, $\tilde{\mathbf{R}} = \mathbf{F} \mathbf{R} \mathbf{F}^{-1}$, the output wavevector becomes identical to the incident wavevector, i.e. $k_x^o = k_x^i$, because the x -axis is kept the same after the reflection (Figure R7c). We choose this transformation to keep track of the transverse wavevector of the single-scattered wave. Since a mirror doesn't add any transverse momentum to the transverse wavevector, the output wavevector is set to be the same as the input wavevector. For this reason, $\tilde{\mathbf{R}}(\mathbf{k}_o, \mathbf{k}_i)$ in Fig. 2e is in the form of a matrix with constant diagonals, while the reviewer's preference will lead to the skewed diagonal as exemplified in the ref. 33. Also, the output phase map in Fig. 2h is flipped with respect to the origin compared to the input phase map in Fig. 2g in our coordinate system.

In the general case when an object function has the spatial frequency component \mathbf{q} , $\mathbf{k}_o = \mathbf{k}_i + \mathbf{q}$ in our transformation while $\mathbf{k}_o = -(\mathbf{k}_i + \mathbf{q})$ in the reviewer's preference. Once again, either choice is fine as long as we are aware of the coordinate system. We think that our choice is intuitive in image reconstruction. In our choice, the momentum change, $\mathbf{k}_o - \mathbf{k}_i$, that the single-scattered wave has experienced is directly related to the spatial frequency spectrum of the object function. The reciprocity is valid because the input from $-(\mathbf{k}_i + \mathbf{q})$ leads to the output $-\mathbf{k}_i$, reciprocal of the input \mathbf{k}_i and output $\mathbf{k}_i + \mathbf{q}$.

We added the detailed coordinate system for our mathematical description to the Supplementary section III.

Figure R7. The choice of the coordinate system. **a**, Incident wave with the transverse wavevector ($k_x^i \neq 0, k_y^i = 0$) to a mirror. **b**, Reflected wave described in the frame where the handedness of the Cartesian coordinate system is preserved. **c**, Reflected wave described at the coordinate system where the transverse wavevector is preserved.

8. Line 353: Title of Figure 3 -> I would say “through a mouse skull” rather than “in a mouse skull”.

This is a good suggestion. We rephrased the caption as advised in the revised manuscript.

Author's Response to Reviewer #3

Yoon et al. describe an imaging method that is essentially a form of aberration-corrected OCT microscopy, where the aberration correction is obtained from a diversity of illumination conditions.

They combine digital and hardware aberration corrections. The power of their method is demonstrated by in vivo imaging of brain tissue through an intact mouse skull. This work shows great promise for various kinds of deep-tissue imaging and the experimental work is very high quality.

We deeply appreciate the reviewer's acknowledging both the performance and importance of our method. We carefully addressed the reviewer's thoughtful comments in our revised manuscript.

In general, the manuscript is well written, however, there seems to be hedging of acronyms. The authors name their current method LS-RMM, however in the results section, the relevant images are indicated as CLASS, which refers to the data processing algorithm.

LS-RMM, as far as I understand from the manuscript text, is a combination of a specific data collection strategy with an enhanced version of the CLASS algorithm. This is more or less clear by going back to the flow diagram in Fig. 2 but I would prefer it to be clearer from the text also.

As the technology is improved over a few generations, it is getting difficult to differentiate the generations without proper naming. Here, we named our new hardware setup as the laser scanning reflection-matrix microscopy (LS-RMM) to make it contrast to the laser scanning confocal microscopy (LSCM). As the reviewer is well aware of, the backbone of our setup is similar to LSCM except that LS-RMM records the amplitude and phase of the non-confocal signals as well as confocal signals. By contrasting LS-RMM with the conventional LSCM, we intended to help biologists to better accommodate our new microscope.

We kept using the term CLASS (closed-loop accumulation of single scattering) algorithm to describe the software part of our work. We consider it a new generic decomposition operation of the reflection matrix which can be recorded by different experimental realizations. The CLASS algorithm decomposes the reflection matrix into $\tilde{\mathbf{R}} = \tilde{\mathbf{P}}_o \tilde{\mathbf{O}} \tilde{\mathbf{P}}_i$ to allow us to separately identify the object function $\tilde{\mathbf{O}}$ and input/output aberration matrices, $\tilde{\mathbf{P}}_i/\tilde{\mathbf{P}}_o$. For comparison, singular value decomposition (SVD) decomposes a reflection (or transmission) matrix into $\tilde{\mathbf{R}} = \mathbf{U} \boldsymbol{\tau} \mathbf{V}^\dagger$ to identify singular values in $\boldsymbol{\tau}$ and the associated eigenchannels in \mathbf{U}/\mathbf{V} . Unless the reviewer has strong objection to using two different acronyms, one for the hardware (LS-RMM) and the other for the software (CLASS), we would like to keep them as intended. We revised our manuscript to make this distinction clear throughout the main text.

The gain of the new method with respect to the previously published CLASS method is significant, but this is only made clear in the supplementary.

We appreciate the reviewer's thorough review of all our manuscript including Supplementary Information and acknowledging the significant gain that we provided in the present study. Indeed, there are important advancements, especially those describing the improvement of the CLASS algorithm dealing with highly space-variant aberrations, that are described in detail in the Supplementary Information. This improvement has been critical for realizing the through-

skull imaging. Frankly speaking, our present study covers too wide a range of advancements — label-free in vivo through-skull imaging, hardware adaptive optics for multi-photon through-skull imaging, and hardware/software improvements that enabled these advancements — to be contained in a single paper. While we are eager to include all the details in the main text, the length limit prevented us from doing so. Instead, we introduced main results in the main text and assigned technical details in the Supplementary Information.

An important point is the relation to recent work from the Boccaro group. A current preprint from that group <https://arxiv.org/abs/1910.07252> introduces quite similar ideas and utilizes essentially the same optical setup to measure through much thicker tissue (but employing brighter objects). The relation to this later work should be discussed, not only the older published work from the same group.

We thank the reviewer for pointing out this reference. The work by the Boccaro group is certainly an interesting approach for deep optical imaging. Their work may appear to be similar to our work in the context of exploiting the time-gated reflection matrix. However, the working principle and effective working regimes are quite different. In their earlier approach (Ref. 32), they made use of singular value decomposition (SVD) of the time-gated reflection matrix to find eigenchannels that preferably couple to bright target objects. In their recent work that the reviewer mentioned, they devised the concept of a distortion matrix, which is the transformation of the output basis to the de-scanned basis by removing the phase ramp added by the beam scanning. Once again, by applying the SVD to the distortion matrix, they identified common phase maps from the sample representing the sample-induced aberrations. SVD is a powerful tool to attenuate the effect of multiple light scattering as we also demonstrated in our earlier study (Jeong, S. *et al.* Focusing of light energy inside a scattering medium by controlling the time-gated multiple light scattering. *Nature Photonics* **12**, 277–283 (2018)), which allows their methods to work for a thick tissue. However, it often requires bright reflecting targets to properly identify the eigenchannels from the target. Furthermore, the eigenchannels are rather indirect measure of the aberrations. For this reason, the SVD approach could resolve the resolution targets of a few micron size, not the diffraction-limited size.

In our group, we have been directly extracting the sample's object function from the measured reflection matrix. In our earlier work in 2015 (Ref. 30), we measured the time-gated reflection matrix for the first time and developed an algorithm termed collective accumulation of single scattering (CASS) that efficiently extracts the object function under strong multiple scattering background. In the following study in 2017 (Ref. 27), we developed the new algorithm termed the closed-loop accumulation of single scattering (CLASS) that addresses multiple scattering and sample-induced aberrations all together. As explained above, the CLASS is a novel operator that decomposes the reflection matrix into $\tilde{\mathbf{R}} = \tilde{\mathbf{P}}_o \tilde{\mathbf{O}} \tilde{\mathbf{P}}_i$. This enables us to directly retrieve the object function $\tilde{\mathbf{O}}$. In comparison, the SVD decomposes a reflection matrix into $\tilde{\mathbf{R}} = \mathbf{U} \boldsymbol{\tau} \mathbf{V}^\dagger$, where $\boldsymbol{\tau}$ contains singular values, not the object function. For this reason, our studies, including the present study, could demonstrate the diffraction-limited imaging of biological samples embedded within biological tissues. Furthermore, we could physically correct the aberrations identified by $\tilde{\mathbf{P}}_i / \tilde{\mathbf{P}}_o$. In the present study in particular, we made the significant improvement of the CLASS algorithm to deal with spatially varying aberrations without losing the spatial frequency resolution.

We added the following sentences to the introduce the SVD approaches including the one that the reviewer mentioned.

“Previous reflection matrix approaches often use singular value decomposition to cope with multiple scattering noise and/or aberrations (Ref.32, 33, 34), which has been specialized for mapping highly reflecting objects with the resolving power of a few microns in the strong scattering regime.”

The final proof by using the recovered aberration map for 2-photon microscopy is strong evidence that the method recovers the true object. Again, it is a pity that this is hidden in the supplementary information, but probably length limitations do not allow otherwise.

In response to the reviewer comment, we added the near-diffraction-limited two-photon fluorescence (TPF) imaging through an intact skull to Fig. 4 of the main text. In our original manuscript, we demonstrated the correction of SHG image at a single isoplanatic patch. In our revision, we prepared a transgenic mouse expressing EGFP at the neuronal membrane and conducted hardware aberration correction of through-skull TPF imaging over multiple isoplanatic patches. As shown in Figure R87, we could map the dendrites and their spines with 500 nm resolution, close to the diffraction limit. As the reviewer remarked, the addition of this new results is a strong evidence that our method recovers the true object.

Figure R8. Two-photon fluorescence imaging through an intact mouse skull. **a**, Conventional OCM image under an intact mouse skull before aberration correction. The thickness of the skull was about 85 μm , and focal plane was set to a depth $z_0 = 125 \mu\text{m}$ from the upper surface of the skull. **b**, LS-RMM image stitched after applying aberration correction to each of 15×15 subregions. **c**, Aberration maps of the subregions indicated by the gray dotted line in **b**. The size of subregion is $10 \times 10 \mu\text{m}^2$, and each phase map contains 9,880 angular modes. Color bar, phase in radians. **d**, TPF image at the same position as **a** before hardware aberration correction. The image was obtained by the maximum intensity projection for a depth range of 119-135 μm with 0.5- μm spacing. **e**, TPF image after physical aberration correction for the subregion indicated by the yellow box in **b**. Yellow boxes in **d** and **e** correspond to the same yellow box area in **b**. **f** and **g**, MIP of TPF images at the depth $z_1 = 113 \pm 1.5 \mu\text{m}$ before and after aberration correction, respectively, for the area indicated as white dashed box in **d**. **h** and **i**, Same as **f** and **g**, respectively, for the depth $z_2 = 122 \pm 1.5 \mu\text{m}$. Color bar, intensity normalized by the maximum intensity in **i**. Scale bars indicate 30 μm in **a**, **b** and **d**, and 10 μm in **e-i**.

In conclusion, this manuscript presents an aberration-corrected OCT method with the potential to image realistic objects through strongly aberrating thin media. I recommend it for publication with attention to the above points.

Once again, we appreciate the reviewer's thoughtful comments and suggestions. Addressing the reviewer's valuable suggestions resulted in the significant improvement of the quality of our study and its scientific integrity.

Reviewer Comments

Reviewer #1 (Remarks to the Author):

The authors have addressed many of the points raised during the initial review. Compared to the earlier work of AO based through skull imaging, the AO method demonstrated in this work has its advantages on using lower laser power and requiring no external label. Regarding to the ground truth images, the authors also added new data (two-photon imaging through 85 micron thin skull). The new data showed that the AO worked nicely to improve the two-photon imaging quality.

A concern is about the in vivo performance. In the in vivo study, the mouse skull was only 85 micron thin. This seems much thinner than some of the earlier work (e.g. 150 micron) or the common adult mouse skull thickness. While the title claims "imaging through intact mouse skull", the actual imaging seems to only work for very thin skull. This seems a bit overclaiming the actual capability of the method.

In addition, for fixed samples, the method seems to be able to correct high order aberration. However, for in vivo imaging, the aberration seems to be mainly low order ones (Fig. 4c).

The label free axon imaging (Fig. 4b) still seems to be of low quality. For example, we can compare with the imaging through cranial window. Schain, Aaron J., Robert A. Hill, and Jaime Grutzendler. "Label-free in vivo imaging of myelinated axons in health and disease with spectral confocal reflectance microscopy." *Nature medicine* 20, no. 4 (2014): 443-449. Although one can argue about the usage of optical window and visible lasers, image quality is very important for practical applications, especially for the myelin pathology studies. With the quality shown in this work, it seems challenging to yield useful information.

Overall, the AO method has its great advantages of using lower laser power and no label. But its AO performance seems to be limited to very thin skulls. The in vivo performance of AO is not as good as that for static samples. The title of "imaging through intact skull" seems overclaiming the actual capability. The axon imaging quality seems poor for practical applications.

Reviewer #2 (Remarks to the Author):

The authors have satisfactorily addressed the points I raised in my initial review.

There is just one point that remains problematic to me: I don't agree with the reply made by the authors to the third reviewer about the recent work from the Boccara group (Ref.33, recently published in *Science Advances*). Unlike the claim made by the authors, the distortion matrix approach introduced in that paper does allow one to reach a diffraction limited resolution (contrary to the former work in Ref.32). The SVD of the distortion matrix directly extracts the corrected

aberration phase laws. Admittedly, the proof-of-concept was made by considering resolution targets as objects to image but this SVD method can also be applied to imaging of more complex objects such as random scattering media (see the work by Lambert et al. recently published in PNAS that applies the same approach in the context of ultrasound imaging). Hence I suggest the authors to rephrase that part since the distortion matrix approach can simultaneously address multiple isoplanatic patches and is thus directly concurrent with the method developed in the present paper.

Once this point will have been clarified, the manuscript will be ready for publication.

Again, congratulations to the authors for this nice piece of work.

Author's Response to Reviewer #1:

The authors have addressed many of the points raised during the initial review. Compared to the earlier work of AO based through skull imaging, the AO method demonstrated in this work has its advantages on using lower laser power and requiring no external label. Regarding to the ground truth images, the authors also added new data (two-photon imaging through 85 micron thin skull). The new data showed that the AO worked nicely to improve the two-photon imaging quality.

We thank the reviewer for acknowledging that our revision has addressed most of his technical concerns. We found that the remaining concerns are rather subjective. In this revision, we addressed them by the most possible objective reasonings.

A concern is about the in vivo performance. In the in vivo study, the mouse skull was only 85 micron thin. This seems much thinner than some of the earlier work (e.g. 150 micron) or the common adult mouse skull thickness. While the title claims "imaging through intact mouse skull", the actual imaging seems to only work for very thin skull. This seems a bit overclaiming the actual capability of the method.

We respectfully disagree with the reviewer's opinion. At first, the thickness of a mouse skull was around 120~150 μm in our in vivo imaging (Fig. 3). More importantly, the skull's thickness varies with individuals and even with locations within the same mouse. Therefore, there is no scientific meaning to specify a particular thickness as a threshold for 'imaging through intact mouse skull.' What matters in imaging is whether to recover object structures that are obscured by the tissues and thus invisible to the existing methods. In our demonstrations in Figs. 3 and 4, the degree of skull's aberration was so severe that even the time-gated confocal imaging didn't show any structures. We made no effort to manipulate the thickness of the skull in all our intact skull imaging and performed imaging through the mouse skull as it were. Therefore, we think that our title accurately describes what was done.

To relieve the reviewer's concern, we introduce an earlier study as an example. The work by Chris Xu's group (Wang, T. *et al.* Three-photon imaging of mouse brain structure and function through the intact skull. *Nature Methods* **15**, 789–792 (2018)) demonstrated multiphoton imaging of vasculature and GCaMP6s calcium imaging through the mouse skull whose thickness was 100~120 μm , similar to our study. The title of the paper contains a phrase, 'through the intact skull.'

In addition, for fixed samples, the method seems to be able to correct high order aberration. However, for in vivo imaging, the aberration seems to be mainly low order ones (Fig. 4c).

The aberration map in Fig. 4c shows less of higher-order aberrations than those in Fig. 3 simply because the skull was slightly thinner. However, this aberration map still contains extremely higher order aberrations that conventional adaptive optics can hardly deal with. Conventional methods based on an iterative feedback by a wavefront shaping device controls typically less than 32×32 segments while our aberration maps consist of 112×112 pixels (~10,000 correction modes within a circular pupil). Therefore, the angular resolution of aberration correction is more than an order of magnitude higher than the conventional methods. According to the theoretical analysis (Ref. 30), the ability to finding aberration maps with such a high resolution is critical. The theoretical study suggests that about 20,000 correction modes are required to

precisely correct for the aberrations induced by an 80- μm -thick skull. Our result goes well with this prediction.

The label free axon imaging (Fig. 4b) still seems to be of low quality. For example, we can compare with the imaging through cranial window. Schain, Aaron J., Robert A. Hill, and Jaime Grutzendler. "Label-free in vivo imaging of myelinated axons in health and disease with spectral confocal reflectance microscopy." Nature medicine 20, no. 4 (2014): 443-449. Although one can argue about the usage of optical window and visible lasers, image quality is very important for practical applications, especially for the myelin pathology studies. With the quality shown in this work, it seems challenging to yield useful information.

As we addressed in our previous revision, the contrast of reflectance imaging of myelinated axons is intrinsically low at the source wavelength of 900 nm. We demonstrated that the contrast of the through-skull image was comparable to that taken in the absence of the mouse skull. Once again, the seemingly low quality image is intrinsic, not due to technical limitations, and this is the price to pay for label-free imaging through an intact skull.

We think that the reviewer underestimates the wavelength effect in comparing our work with Grutzendler group's work (Ref. 40). We obtained a confocal image at visible wavelength and compared it with the confocal image taken at 900 nm for the skull-free brain tissues (Fig. R1). This result clearly shows that visible wavelength is far more beneficial in terms of image contrast. However, there is again a price to pay for this better contrast. The skull-induced aberration is much more severe in the visible wavelength, making the through-skull imaging extremely challenging.

Figure R1. Comparison of reflectance images with different wavelengths. The left-hand image is an in vivo image of myelinated axons in mouse brain through a cranial window at 70- μm depth under the dura taken using 633 nm wavelength. The right image is an OCM image taken at the surface of a fixed brain tissue using 900 nm wavelength.

The reviewer seems to narrow down the applicability of imaging myelinated axons to such a case shown by Grutzendler group. Certainly, visible wavelength confocal imaging visualizing the individual myelin internodes can provide details required for some of the myelin pathology studies. However, there are many applications of myelin imaging that do not require mapping of the internodes. In our study, we could recover the images of the myelinated axons that were invisible to the conventional methods. Individual myelinated fibers were clearly resolved, and the image quality was good enough to ascertain the formation of myelin. Our method can

potentially be applied to those studies based on the similar image quality as ours, but through intact skull. Exemplary studies are listed below.

- A quantitative analysis of myelinated axons such as the evaluation of pathological demyelinating disease (V. J. Srinivasan, et. al., “Optical coherence microscopy for deep tissue imaging of the cerebral cortex with intrinsic contrast”, *Opt. Express*, v. 20, 2220-2239 (2012)),
- Non-invasive high-resolution delineation of brain tumor by virtue of the destruction of myelin at the tumor site (V.-H. Le, et. al., “Brain tumor delineation enhanced by moxifloxacin-based two-photon/CARS combined microscopy”, *Biomedical Optics Express*, 8(4), 2148–2161 (2017)),
- Study on temporal or regional development of cortical myelin, or optical tractography to investigate structural connections of myelinated axons for neurological disorder studies (H. Wang, et. al., “Reconstructing micrometer-scale fiber pathways in the brain: Multi-contrast optical coherence tomography based tractography”, *Neuroimage*, 58, (2011), pp. 984-992).

As an additional technical note, the myelinated fibers may appear less continuous in Fig. 4b. We attribute this to large inclination angles of the fibers with respect to the imaging plane. The image shows a maximum-intensity projection (MIP) of only five images taken at 4- μm axial interval. The 4- μm axial spacing was rather coarse to optimally visualize the myelinated fibers running through the image plane with stiff angles. However, this axial sampling was good enough to find aberration maps required for the hardware wavefront correction of the two-photon fluorescence (TPF) imaging of dendrites and spines, which was the main purpose of the demonstration in Fig. 4. The MIP images of the TPF imaging (Figs. 4g and 4i) show continuous fibrous structures of dendrites as the depth scanning step was 1.5 μm . If we were interested in label-free imaging in Fig. 4, we would have taken finer axial scanning steps. We added the following sentences to the caption of Fig. 4b.

“MIP of five LS-RMM images taken over a depth range of 117-133 μm with 4- μm steps. LS-RMM image for each depth was obtained by stitching aberration-corrected 15 \times 15 subregions. Note that myelinated fibers appear rather discontinuous in the image mainly due to coarse depth scanning steps and large inclination angles of the fibers with respect to the image plane.”

Overall, the AO method has its great advantages of using lower laser power and no label. But its AO performance seems to be limited to very thin skulls. The in vivo performance of AO is not as good as that for static samples. The title of “imaging through intact skull” seems overclaiming the actual capability. The axon imaging quality seems poor for practical applications.

Once again, we appreciate the reviewer’s critical opinions on the outcome of our study. We’d like to emphasize that the present work is the first study demonstrating in vivo label-free imaging of myelinated axons through an intact mouse skull, and we believe that this claim has been faithfully supported by experimental data. However, this doesn’t mean that there is no room to improve. The image quality of in vivo imaging is a bit lower than ex vivo imaging due to the motion artifacts. We are planning to improve the imaging speed by using a high-speed camera and devise an improved algorithm dealing with motion artifacts. At the same time, we are currently working on imaging through an intact skull at visible wavelengths to obtain high-contrast images. This study will require further improvement in the aberration correction

algorithm to deal with the increased scattering and aberration. We added the following sentences to expand the future outlook of our study.

“LS-RMM can employ any excitation wavelength, similar to confocal reflectance imaging. The use of longer excitation wavelengths can potentially be helpful as the multiple scattering noise is to be reduced³⁹. The use of visible wavelengths will lead to increasing the image contrast and spatial resolution, which will open the possibility of investigating detailed myelin pathologies through intact skull⁴⁰.”

Author’s Response to Reviewer #2

The authors have satisfactorily addressed the points I raised in my initial review.

We are happy to know that the reviewer is satisfied with our previous revision.

There is just one point that remains problematic to me: I don't agree with the reply made by the authors to the third reviewer about the recent work from the Boccara group (Ref.33, recently published in Science Advances). Unlike the claim made by the authors, the distortion matrix approach introduced in that paper does allow one to reach a diffraction limited resolution (contrary to the former work in Ref.32). The SVD of the distortion matrix directly extracts the corrected aberration phase laws. Admittedly, the proof-of-concept was made by considering resolution targets as objects to image but this SVD method can also be applied to imaging of more complex objects such as random scattering media (see the work by Lambert et al. recently published in PNAS that applies the same approach in the context of ultrasound imaging). Hence I suggest the authors to rephrase that part since the distortion matrix approach can simultaneously address multiple isoplanatic patches and is thus directly concurrent with the method developed in the present paper.

Once this point will have been clarified, the manuscript will be ready for publication.

Again, congratulations to the authors for this nice piece of work.

As we discussed in our response to the third reviewer, the distortion matrix concept is an interesting method taking different approach from LS-RMM. We acknowledge that our statement therein was solely based on what was demonstrated in Ref. 33, which is now replaced with the recently published Science Advances paper of the same work, in the context of optical imaging. Since ultrasound imaging employing the same method demonstrated the diffraction-limited imaging in complex media (Lambert, W., Cobus, L. A., Frappart, T., Fink, M. & Aubry, A. Distortion matrix approach for ultrasound imaging of random scattering media. *Proc National Acad Sci* **117**, 14645–14656 (2020)) as the reviewer mentioned, we expect that similar demonstration can be made in optical imaging in the future study.

We revised the text introducing the Science Advances paper (Ref. 34 in this revision) as follows and added the PNAS paper as Ref. 36.

“Previous reflection matrix approaches often use singular value decomposition to cope with multiple scattering noise and/or aberrations^{33–36}. In the context of optical imaging, it was demonstrated that a sharp image was recovered for a few-micron-sized highly reflecting resolution target hidden under a scattering and aberrating tissue³⁴.”